# THRESHOLD-CONSISTENT MARGIN LOSS FOR OPEN-WORLD DEEP METRIC LEARNING

**Qin Zhang**[1]*, **Linghan Xu**[1]*, **Qingming Tang**[2], **Jun Fang**[1], **Ying Nian Wu**[1], **Joe Tighe**[1], **Yifan Xing**[1]
[1] AWS AI Labs, [2] Alexa AI
{qzaamz, linghax, qmtang, junfa, wunyin, yifax}@amazon.com, jtighe@cs.unc.edu

## ABSTRACT

Existing losses used in deep metric learning (DML) for image retrieval often lead to highly non-uniform intra-class and inter-class representation structures across test classes and data distributions. When combined with the common practice of using a fixed threshold to declare a match, this gives rise to significant performance variations in terms of false accept rate (FAR) and false reject rate (FRR) across test classes and data distributions. We define this issue in DML as **threshold inconsistency**. In real-world applications, such inconsistency often complicates the threshold selection process when deploying commercial image retrieval systems. To measure this inconsistency, we propose a novel variance-based metric called **O**perating-**P**oint-**I**nconsistency-**S**core (OPIS) that quantifies the variance in the operating characteristics across classes. Using the OPIS metric, we find that achieving high accuracy levels in a DML model does not automatically guarantee threshold consistency. In fact, our investigation reveals a Pareto frontier in the high-accuracy regime, where existing methods to improve accuracy often lead to degradation in threshold consistency. To address this trade-off, we introduce the **T**hreshold-**C**onsistent **M**argin (TCM) loss, a simple yet effective regularization technique that promotes uniformity in representation structures across classes by selectively penalizing hard sample pairs. Extensive experiments demonstrate TCM's effectiveness in enhancing threshold consistency while preserving accuracy, simplifying the threshold selection process in practical DML settings.

## 1 INTRODUCTION

Deep metric learning (DML) has shown success in various open-world recognition and retrieval tasks (Schroff et al., 2015a; Wu et al., 2017; Deng et al., 2019; Wang et al., 2018). Nevertheless, the common DML losses, such as contrastive loss (van den Oord et al., 2018; Chen et al., 2020), pairwise loss (Brown et al., 2020; Patel et al., 2022) and proxy-based losses (Kim et al., 2020; Movshovitz-Attias et al., 2017; Qian et al., 2019; Deng et al., 2019), often yield highly varied intra-class and inter-class representation structures across classes (Liu et al., 2019; Duan et al., 2019; Zhao et al., 2019). Hence, even if an embedding model has strong separability, distinct classes may still require varying thresholds to uphold a consistent operating point in terms of false reject rate (FRR) or false acceptance rate (FAR). This challenge is particularly important in real-world image retrieval systems, where a threshold-based retrieval criterion is preferred over a top-k approach due to its ability to identify negative queries without matches in the gallery. However, selecting the right threshold is difficult, especially when systems must cater to diverse use-cases. For instance, in clothing image retrieval for online shopping, the similarity between two T-shirts can be significantly different from that between two coats. A threshold that works well for coats may lead to poor relevancy and give many false positives in the retrieved images for T-shirts, as shown in Figure 1. These difficulties are more pronounced in the ***open-world*** scenarios (Scheirer et al., 2012; Bendale & Boult, 2015; 2016), where the test classes may include entirely new classes not seen during training.

We define the phenomenon in DML, where different test classes and distributions require varying distance thresholds to achieve a similar retrieval or recognition accuracy, as ***threshold inconsistency***. In commercial environments, particularly under the practical evaluation and deployment

---

*Equal contribution.

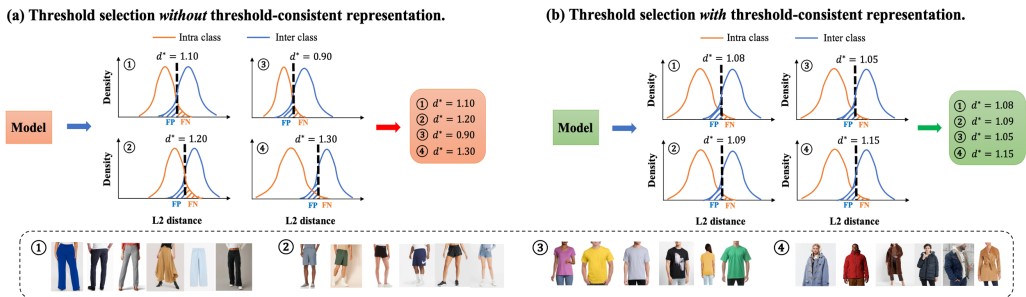

Figure 1: Here we show that (a) without threshold-consistent representation, selecting the right threshold for a commercial image retrieval system that serves a diverse range of test classes and distributions is challenging. It requires careful manual tuning of retrieval thresholds to strike a balance across multiple datasets. However, (b) with threshold-consistent representation, different test distributions yield similar distance thresholds at the performance target, effectively simplifying the otherwise complicated manual threshold tuning process. In the plots, $d^*$ denotes the distance threshold selected to align the False Positive (FP) rate with a pre-defined target.

setting with one fixed threshold for diverse user groups (Liu et al., 2022), the significance of threshold inconsistency cannot be overstated. Accurate quantification of this inconsistency is essential for detecting potential biases in the chosen threshold. To this end, we introduce a novel evaluation metric, named **O**perating-**P**oint-**I**nconsistency-**S**core (OPIS), which quantifies the variance in the operating characteristics across classes within a target performance range. Using OPIS, we observe an accuracy-threshold consistency Pareto frontier in the high accuracy regime, where methods to improve accuracy often result in a degradation in threshold consistency, as shown in Figure 3. This highlights that ***achieving high accuracy does not inherently guarantee threshold consistency***.

One solution to this problem is using posthoc calibration methods (Platt et al., 1999; Zadrozny & Elkan, 2002; Guo et al., 2017a), which adjust a trained model's distance thresholds to align with specific operating points in FAR or FRR. However, in real-world settings, these methods can be inefficient and lack robustness, as they involve constructing separate calibration datasets and may require prior knowledge about the test distribution for effective calibration (Naeini et al., 2015; Guo et al., 2017a). Moreover, they do not address the threshold inconsistency problem unless customized calibration is done for each user. Another option is employing conformal prediction (Romano et al., 2020; Gibbs & Candes, 2021), which guarantees confidence probability coverage and can handle complex data distributions as well as covariate and label shifts. However, conformal prediction inherently assumes a closed-world setting, where training and test samples share the same label space. In contrast, real-world image retrieval systems typically operate in an open-world environment, presenting a more complex and realistic setting with unknown classes at test time.

Given these challenges, an essential question arises: Can we train an embedding model for open-world image retrieval that sustains a consistent distance threshold across diverse data distributions, thus avoiding the complexities of posthoc threshold calibration? This objective falls within the scope of calibration-aware training. In closed-set classification, the goal of calibration-aware training is to align predicted confidence probabilities with empirical correctness of the model (Guo et al., 2017a; Müller et al., 2019; Mukhoti et al., 2020). However, our focus lies on what we term as ***threshold-consistent DML***, a paradigm that trains an embedding model with reduced threshold inconsistencies, such that a universal distance threshold can be applied to different test distributions to attain a similar level of FAR or FRR. This differentiation is crucial because in DML the output similarity score does not strictly reflect the empirical correctness of the model (Xu et al., 2023) and may exhibit strong variations across test data distributions. To address the unique challenges of threshold inconsistency in DML, we propose a simple yet effective regularization technique called **T**hreshold-**C**onsistent **M**argin (TCM) loss. Through experiments on four standard image retrieval benchmarks, we validate the efficacy of the TCM regularization in improving threshold consistency while maintaining accuracy. To summarize, our contributions are as follows:

- We propose a novel variance-based metric, named **O**perating-**P**oint-**I**nconsistency-**S**core (OPIS), to quantify the threshold inconsistency of a DML model. Notably, OPIS does not need a separate hold-out dataset besides the test set, enhancing flexibility in evaluation.

- We observe an accuracy-threshold consistency Pareto frontier in the high accuracy regime. This finding underscores that achieving high model accuracy in DML does not automatically guarantee threshold consistency, necessitating dedicated solutions.

- We introduce the **T**hreshold-**C**onsistent **M**argin (TCM) loss, a simple yet effective regularization technique, that can be combined with any base losses and backbone architecture to improve threshold consistency in DML. Our approach outperforms SOTA methods across various standard image retrieval benchmarks, demonstrating substantial improvements in threshold consistency while maintaining or even enhancing accuracy.

## 2 RELATED WORKS

**DML losses for image retrieval** Advancements in DML losses for image retrieval have focused on improving accuracy, scalability and generalization Brown et al. (2020); Patel et al. (2022); Deng et al. (2020); Kim et al. (2023); Roth et al. (2020); Kan et al. (2022); Ypsilantis et al. (2023). The pioneering work of the *Smooth-AP* loss (Brown et al., 2020) optimizes a smoothed approximation for the average precision. Similarly, the *Recall@k Surrogate* loss (Patel et al., 2022) approximates the *recall@k* metric. Leveraging vision-transformer backbones and large batch sizes, *Recall@k Surrogate* has achieved remarkable performance in several image retrieval benchmarks. However, these pairwise methods are inefficient when dealing with a large number of classes. To reduce the computational complexity, proxy-based methods such as *ProxyAnchor* (Kim et al., 2020), *ProxyNCA* (Movshovitz-Attias et al., 2017), *SoftTriple* (Qian et al., 2019), *ArcFace* (Deng et al., 2019), and *HIER* (Kim et al., 2023) are employed, where sample representations are compared against class prototypes. Despite high accuracy, these methods still face challenges in biases and fairness (Fang et al., 2013; Ilvento, 2019; Dullerud et al., 2022) and display inconsistencies in distance thresholds when applied in real-world scenarios (Liu et al., 2022).

**Evaluation Metrics for threshold consistency (inconsistency)** In closed-set classification, threshold consistency is usually evaluated through calibration metrics, such as Expected Calibration Error (ECE) (Naeini et al., 2015), Maximum Calibration Error (MCE) (Guo et al., 2017b) and Adaptive ECE (Nixon et al., 2019). These metrics gauge how well a model's predictions match actual correctness. However, directly applying them to evaluate threshold consistency in DML (e.g., by replacing confidence probability with similarity measures) is not straightforward. A key hurdle is that DML uses distance measurements to represent semantic similarities, and these distances can vary widely across different classes due to the intrinsic non-bijectiveness of semantic similarity in the data (Roth et al., 2022). In the context of DML, OneFace (Liu et al., 2022) introduced the *calibration threshold* for face recognition systems, which corresponds to the distance threshold at a given FAR of a separate calibration dataset. They further propose the One-Threshold-for-All (OTA) evaluation protocol to measure the difference in the accuracy performance across datasets at this calibration threshold as an indicator for threshold consistency. However, this approach requires a dedicated calibration dataset, which can be difficult to acquire in practice. To our knowledge, there is no widely accepted and straightforward metric for threshold consistency in DML.

**Calibration-aware training vs Posthoc threshold calibration** Calibration-aware training has been well studied in closet-set classification, where the goal is to align predicted probabilities with empirical correctness (Guo et al., 2017a; Müller et al., 2019; Mukhoti et al., 2020). Common approaches use a regularizer to guide the model in generating more calibrated predictions (Pereyra et al., 2017; Liang et al., 2020; Hebbalaguppe et al., 2022). Yet, *threshold-consistent training for DML differs from calibration-aware training*. Instead of aligning model output with empirical correctness, threshold-consistent DML seeks to maintain a consistent distance threshold across classes and data distributions. In face recognition, Liu et al. (2022) introduces the Threshold Consistency Penalty to improve threshold consistency among various face domains. The method divides mini-batch data into 8 domains and computes each domain threshold using a large set of negative pairs from a feature queue. It then adjusts the loss contribution from each sample based on the ratio of its domain threshold to the in-batch calibration threshold. However, this method is designed for face recognition – a more constrained scenario. In contrast, our target is general image retrieval tasks which can involve significantly more domains, making it impractical to construct negative pairs for all domains. Besides train-time methods, another approach is posthoc threshold calibration, such as Platt calibration (Platt et al., 1999), isotonic regression (Zadrozny & Elkan, 2002) and temperature scaling (Guo et al., 2017a), which seeks to calibrate the operating point of a trained model using

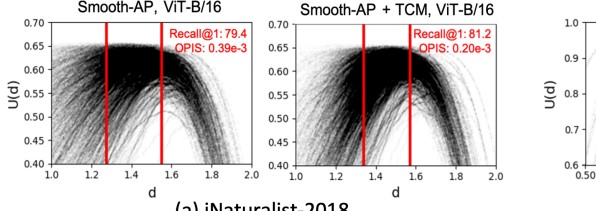 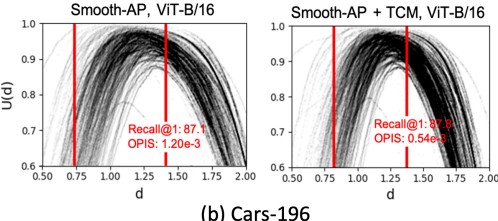

(a) iNaturalist-2018 (b) Cars-196

Figure 2: *Utility - Distance Threshold* Curves are presented for test classes in the iNaturalist-2018 and Cars-196 datasets. Each curve represents a class in its respective dataset. The calibration range is underscored by the red lines. As defined in Eq.2, the calibration range is based on a pre-defined FAR or FRR target. Thus, changing the loss can result in minor shifts in the calibration range. After integration of the TCM regularization, a significant enhancement in the alignment of utility curves across various classes is observed, accompanied by a notable enhancement in threshold consistency, as indicated by the reduction in OPIS by up to 55%.

hold-out calibration datasets. However, it cannot solve threshold inconsistency unless customized calibration is conducted for each user. Another category of posthoc calibration method is conformal prediction (Tibshirani et al., 2019; Romano et al., 2020; Gibbs & Candes, 2021; Barber et al., 2023), which can be applied beyond the setting of exchangeable data even when the training and test data are drawn from different distributions. However, conformal prediction relies on a closed-set setting where the training and test data share the same label space, which does not apply to open-world image retrieval. Thus, in this work, we focus on developing a threshold-consistent training technique tailored for DML, with the goal of simplifying the posthoc calibration process in practical settings.

## 3 THRESHOLD INCONSISTENCY IN DEEP METRIC LEARNING

**Visualizing threshold inconsistency in image retrieval** We visually illustrate the issue of threshold inconsistency in DML using image retrieval datasets. First, we borrow the widely-used $F$-score (Sasaki et al., 2007) to define the utility score, incorporating both sides of the accuracy metric (e.g. precision and recall, or specificity and sensitivity). Specifically, we denote one side as $\phi$ and the other side as $\psi$, and define the utility score, denoted as U, as follows:

$$\text{U}(d) = \frac{(1 + c^2) \cdot \phi(d) \cdot \psi(d)}{c^2 \phi(d) + \psi(d)} \tag{1}$$

where $d$ is the distance threshold ($d \in [0, 2]$ for hyperspherical embeddings), and $c$ is the relative importance of $\psi$ over $\phi$ ($c = 1$ if not specified). Without loss of generality, we let $\phi$ be specificity (same as TNR or $1 - $ FAR) and $\psi$ be sensitivity (same as recall or $1 - $ FRR).[1]

In Figure 2, we present the *accuracy utility-distance threshold* curves for the test classes using models trained on the iNaturalist-2018 (Horn et al., 2017) and Cars-196 (Krause et al., 2013) datasets. In the left column of each subfigure, we observe considerable variations in the operating characteristics among distinct classes for models trained with the popular *Smooth-AP* loss. These variations make it difficult to select a single distance threshold that works well across the entire spectrum of test distributions. However, while we will elaborate on in later sections, incorporating our proposed TCM regularization during training visibly improves the threshold consistency across classes, as evidenced by the more aligned utility curves compared to those without the TCM regularization.

**OPIS for overall threshold inconsistency** To quantify threshold inconsistency in DML, we introduce a variance-based metric, **O**perating-**P**oint-**I**nconsistency **S**core (OPIS). Unlike the OTA evaluation proposed in Liu et al. (2022), OPIS does not require a separate calibration dataset. It quantifies the variance in the operating characteristics across test classes in a predefined *calibration range* of distance thresholds. This calibration range, denoted as $[d_{\min}, d_{\max}]$, is typically determined based on the target performance metric operating ranges (e.g., $a <$FAR$< b$, where $a, b$ are pre-determined error constraints). Formally, the OPIS metric can be expressed as follows:

$$\text{OPIS} = \frac{\sum_{i=1}^{T} \int_{d_{\min}}^{d_{\max}} ||\text{U}_i(d) - \bar{\text{U}}(d)||^2 \, \mathrm{d}d}{T \cdot (d_{\max} - d_{\min})} \tag{2}$$

---

[1]We employ the specificity and sensitivity pair because they are particularly relevant for visual recognition applications and are not sensitive to changes in test data composition.

where $i = 1, 2, ..., T$ is the index for the test classes, $U_i(d)$ is the accuracy utility for class $i$, and $\bar{U}(d)$ is the average utility for the entire test dataset.

$\epsilon$**-OPIS for utility divide between groups** The overall OPIS metric does not emphasize on the outlier classes. For applications where outlier threshold consistency is essential, we also provide a more fine-grained metric that focuses on the utility disparity between the best and worst sub-groups. First, we define the utility of the $\varepsilon$ percentile of best-performing classes as follows:

$$U_{\varepsilon_{\text{best}}}(d) = \frac{\phi_{\varepsilon_{\text{best}}}(d) \cdot \psi_{\varepsilon_{\text{best}}}(d)}{\phi_{\varepsilon_{\text{best}}}(d) + \psi_{\varepsilon_{\text{best}}}(d)} \tag{3}$$

where $\phi_{\varepsilon_{\text{best}}}(d)$, $\psi_{\varepsilon_{\text{best}}}(d)$ are the expected accuracy metrics for the entirety of the $\varepsilon$ percentile of the best-performing classes. By replacing $\varepsilon_{\text{best}}$ with $\varepsilon_{\text{worst}}$, the same can be defined for $U_{\varepsilon_{\text{worst}}}(d)$. Then, we define the $\varepsilon$-OPIS metric as the following:

$$\varepsilon\text{-OPIS} = \frac{1}{d_{\max} - d_{\min}} \int_{d_{\min}}^{d_{\max}} ||U_{\varepsilon_{\text{worst}}}(d) - U_{\varepsilon_{\text{best}}}(d)||^2 \, \mathrm{d}d \tag{4}$$

By definition, the $\varepsilon$-OPIS metric is maximized at $\varepsilon \to 0$, and eventually becomes zero when $\varepsilon \to 100\%$ as the best-performing set and worst-performing set become identical.

**High accuracy $\neq$ High threshold consistency** In Figure 3, we employ the OPIS metric to examine the relations between threshold inconsistency and recognition error in embedding models trained with various DML losses, backbones and batch sizes. Notably, we observe distinct behaviors across different accuracy regimes. In the low-accuracy regime, located in the right of the plot, we notice a simultaneous improvement of accuracy and threshold consistency. This aligns with the established notion that improving model discriminability helps threshold consistency by strengthening the association between samples and their corresponding class centroids. However, as the error decreases, a trade-off surfaces in the high-accuracy regime. Here, the reduction in error is correlated with increased threshold inconsistency, leading to the formation of a Pareto frontier.

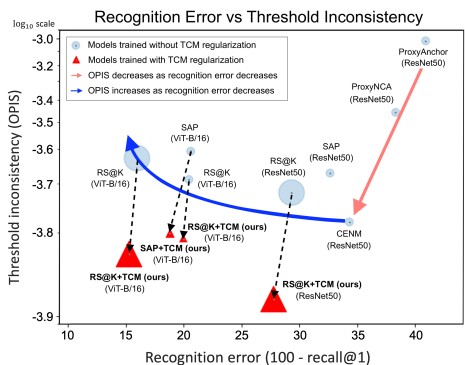

Figure 3: The plot depicts the relations between recognition error (measured by $100 - \text{recall}@1$, the lower the better) and threshold inconsistency (measured by OPIS, the lower the better) across low- and high-accuracy regimes. Each ● represents a trained DML model, with its size indicating the batch size used during training. In the low accuracy regime, located in the right side of the plot, there is a simultaneous improvement in threshold consistency and accuracy, as highlighted by ✓. However, beyond a certain point, a Pareto frontier emerges (indicated by ↘), where enhancing accuracy comes at the expense of threshold consistency. Notably, the inclusion of our proposed TCM regularization (marked in ▲) leads to a substantial OPIS reduction, well below the marked Pareto frontier. *Best viewed in color.*

The trade-off between recognition error and threshold inconsistency highlights that achieving high accuracy alone does not automatically guarantee threshold consistency. In this context, introducing the proposed OPIS metric as an additional evaluation criterion alongside recall@k is crucial for threshold-based commercial DML applications, where the ability to identify negative queries without matching classes in the gallery is of importance. To explain further, we compare OPIS with the widely-used accuracy metric, *recall@k*. These two metrics evaluate different aspects of a model and can be used complementarily: *recall@k* focuses on top-k relevancy (retrieving top-k similar samples as the query from a collection), and OPIS measures the inconsistency in threshold-relevancy (retrieving similar examples above a threshold from a collection). Moreover, unlike *recall@k* that solely gauges recall, OPIS evaluates both the FAR and FRR (=recall), offering a more holistic error assessment.

## 4 TOWARDS THRESHOLD-CONSISTENT DEEP METRIC LEARNING

To tackle the threshold inconsistency problem, we introduce the Threshold-Consistent Margin (TCM) loss. TCM specifically penalizes hard positive and hard negative sample pairs near the decision boundaries outlined by a pair of cosine margins. This strategy is in line with several studies (Dong et al., 2017; Xuan et al., 2020; Robinson et al., 2020) that emphasize hard mining for

Figure 4: (a) An overview of the threshold-consistent DML training framework. Here, the base loss and TCM regularization are combined in an additive fashion to reduce the trade-off between accuracy and threshold consistency. (b) Distinguishing TCM from Margin-Based Softmax Loss such as Deng et al. (2019). See **TCM vs Margin-based Softmax loss** section for detailed explanation of the differences. In this illustration, $\theta_1$ and $\theta_2$ represent the intra-class arc lengths for the blue and red classes, respectively. $x_1$ and $x_3$ are both instances of the blue class with class centroid $W_1$, whereas $x_2$ belongs to the red class with centroid $W_2$. In this case, $x_1, x_2$ are hard negative sample pairs, and $x_1, x_3$ are hard positive sample pairs. *Best viewed in color.*

extracting more informative samples. Let $S^+$ and $S^-$ be the sets of cosine similarity scores for positive and negative pairs in a mini-batch, respectively, the TCM loss is formulated as follows:

$$L_{\text{TCM}} = \lambda^+ \cdot \frac{\sum_{s \in S^+} (m^+ - s) \cdot \mathbf{1}_{s \leq m^+}}{\sum_{s \in S^+} \mathbf{1}_{s \leq m^+}} + \lambda^- \cdot \frac{\sum_{s \in S^-} (s - m^-) \cdot \mathbf{1}_{s \geq m^-}}{\sum_{s \in S^-} \mathbf{1}_{s \geq m^-}} \quad (5)$$

where $\mathbf{1}_{\text{condition}} = 1$ if the condition is true, and $0$ otherwise. $\lambda^+$ and $\lambda^-$ are the weights assigned to the positive and negative regularizations, respectively. The TCM regularizer can be combined with any base loss $L_{\text{base}}$, resulting in the final objective function:

$$L_{\text{final}} = L_{\text{base}} + L_{\text{TCM}} \quad (6)$$

**Design justification: representation structures** Several works have shown a strong correlation between model accuracy and representation structures (Yu et al., 2020; Chan et al., 2022). Indeed, SOTA DML losses are designed to optimize this relationship by encouraging intra-class compactness and inter-class discrimination. However, when considering threshold consistency, the focus shifts towards achieving consistent performance in FAR and FRR in the *calibration range*, with an emphasis on local representation structures near the distance threshold. In this context, the TCM regularization serves as a "local inspector" by selectively adjusting hard samples to prevent over separateness and excessive compactness in the vicinity of the margin boundaries. This strategy also aligns with previous work that found excessive feature compression actually hurts DML generalization (Roth et al., 2020). Since the margin constraints are applied globally, this helps encourage more equidistant distribution of class centroids and more uniform representation compactness across different classes in the embedding space.

**Hard mining strategy** TCM regularizes on hard samples, distinguishing it from techniques that encourage similarity consistency by minimizing marginal variance (Kan et al., 2022). Specifically, TCM's hard mining strategy is different from the semi-hard negative mining strategy (Schroff et al., 2015b) and its variants (Oh Song et al., 2016; Wu et al., 2017; Wang et al., 2019), as TCM's hard mining is based on the absolute cosine similarity values, rather than their relative differences. Meanwhile, TCM also differs from ROADMAP (Ramzi et al., 2021) in that TCM utilizes hard positive and negative counts, whereas ROADMAP uses the total positive and negative counts. This makes TCM well-suited for scenarios involving large batch sizes (as is the standard in DML) and significant imbalances between the counts of positive and negative pairs[2].

**Connection to the calibration range** TCM is implicitly connected to the calibration range of the OPIS metric through the two cosine margins. Since cosine similarity is bijective with the $L_2$ distance for hyperspherical embeddings, these margin constraints ensure that the model's intra-class and inter-class representation structures adhere to the desired distance threshold range, which is $[\sqrt{2 - 2m^+}, \sqrt{2 - 2m^-}]$. However, due to the inevitable distributional shift between the training and testing datasets, the selection of the margin constraints requires some hyper-parameter tuning and cannot be directly estimated from the calibration range. In Figure 6, we give guidance on how to select the margins, with details discussed in the ablation of TCM margin hyperparameters.

**TCM vs Margin-based Softmax loss** TCM has distinguishing characteristics when compared to margin-based softmax losses (Deng et al., 2019; Qian et al., 2019), as illustrated in Figure 4(b).

---

[2]An detailed comparison between TCM and the method of Ramzi et al. (2021) is given in appendix A.2.2

First, TCM is designed as a regularizer that operates in conjunction with a base loss. It specifically applies to hard sample pairs that are located near the margin boundaries. Second, TCM employs two cosine margins to regularize the intra-class and inter-class distance distributions simultaneously. This allows TCM to capture both hard positive and hard negative examples, resulting in more hard pairs within a mini-batch. Secondly, the TCM loss is specifically applied to the hard pairs, contrasting with *Arcface*, which is applied to all pairs. Last, TCM is a sample-level pair-wise loss, which better models the relationships between individual samples compared to proxy-based methods.

**Visualization of TCM effect** We visualize the effect of the TCM regularization on representation structures across the 10 classes in the MNIST dataset of handwritten digits (LeCun et al., 1998) by training a shallow CNN using the *Arcface* loss (Deng et al., 2019). For clearer visualization, we use two-dimensional features and employ kernel-density estimation (Chen, 2017) to model the probability density function for the embeddings of each class. As shown in Figure 5, compared to using *ArcFace* (Deng et al., 2019) only, the incorporation of TCM (*ArcFace*+TCM) enhances the separation between digits 2 and 5 (lower middle), 0 and 8 (lower right), and 4 and 9 (upper left). This observation supports our claims about TCM's ability in refining the representation structures for improved threshold consistency.

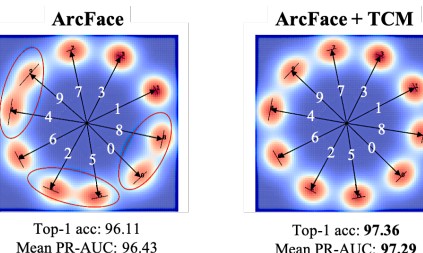

Figure 5: We present visualizations of 2D embedding distributions for the MNIST dataset, both with and without TCM regularization. In the figure, each arrow's direction corresponds to a class centroid and is labeled with its respective digit in white. The width of the line perpendicular to each arrow reflects the intra-class representation compactness, with narrower lines indicating more compact embeddings. The color intensity conveys the probability density distribution of embeddings within each class, with higher density depicted in red.

## 5 EXPERIMENTS

### 5.1 DATASETS AND IMPLEMENTATION DETAILS

**Datasets** For training and evaluation, we use four commonly-used image retrieval benchmarks, namely iNaturalist-2018 (Horn et al., 2017), Stanford Online Product (Song et al., 2015), CUB-200-2011 (Wah et al., 2011) and Cars-196 (Krause et al., 2013). These benchmarks cover a diverse set of data domains including natural species, online catalog images, birds, and cars. As in previous works (Brown et al., 2020; Patel et al., 2022; An et al., 2023), the iNaturalist and Stanford Online Product datasets use an open-world train-test-split, where the training classes are disjoint from the ones in testing. For CUB and Cars, we use shared train-test classes to make fair comparisons with prior DML methods[3]. The details to each dataset can be found in Table 1.

**Evaluation metrics** We measure model accuracy using the *recall@k* metric and assess threshold inconsistency using the OPIS and $\epsilon$-OPIS metrics as defined earlier. Similar to previous works (Veit & Wilber, 2020; Liu et al., 2022), we estimate threshold inconsistency by comparing normalized features of image pairs in 1:1 comparisons. In the case of the iNaturalist-2018 and Stanford Online Product datasets, given the large number of classes, we only sample positive pairs exhaustively and randomly sample negative pairs with a fixed negative-to-positive ratio of 10-to-1 for each class. All positive and negative pairs in the CUB and Cars datasets are exhaustively sampled.

**Implementation details** We use two backbone architectures, namely ResNet (He et al., 2016) and Vision Transformer (Dosovitskiy et al., 2020), both pretrained on ImageNet[4]. Since the original papers do not report OPIS, we train both baseline models (without TCM) and TCM-regularized models using the same configuration. The hyperparameters for each base loss are taken from the original papers. For TCM, we set $\lambda^+ = \lambda^- = 1$. For OPIS, the calibration range is set to 1e-2 $\leq$ FAR $\leq$ 1e-1 for all benchmarks. The margin parameters ($m^+$, $m^-$) are tuned using grid search on 10% of the training data for each benchmark. We adopt the same optimization schemes as specified

---

[3]We also provide results for CUB and Cars in the open-world setting in Appendix A.2.5.

[4]For ResNet, we follow Brown et al. (2020) and use ImageNet-pretrained backbones. For ViTs, we follow Patel et al. (2022) and use ImageNet-21k pretrained backbones released by timm library (Wightman, 2019).

Table 1: Dataset statistics. The datasets with a open-world train-test split are highlighted in light gray.

| Dataset | Train | | | Test | | |
|---|---|---|---|---|---|---|
| | # Ims | # Cls | # Ims/Cl | # Ims | # Cls | # Im/Cl |
| iNat | 325846 | 5,690 | 57.3 | 136093 | 2,452 | 55.5 |
| SOP | 59551 | 11318 | 5.3 | 60502 | 11316 | 5.3 |
| CUB | 5994 | 200 | 30.0 | 5794 | 200 | 29.0 |
| Cars | 8054 | 196 | 41.1 | 8131 | 196 | 41.5 |

Table 2: The influence of TCM regularization on different base losses for $ResNet50^{512}$ backbones.

| $L_{base} + L_{reg}$ | R@1 | R@4 | R@16 | OPIS×1e-3 |
|---|---|---|---|---|
| ProxyNCA + TCM | 63.1 (↑1.4) | 78.6 (↑1.4) | 88.3 (↑1.0) | 0.38 (↓0.16) |
| ArcFace + TCM | 63.6 (↑1.0) | 78.6 (↑0.9) | 88.3 (↑0.8) | 0.25 (↓0.05) |
| SAP + TCM | 69.1 (↑1.7) | 82.9 (↑1.0) | 91.1 (↑0.7) | 0.17 (↓0.16) |
| RS@K + TCM | 72.2 (↑1.5) | 84.9 (↑1.2) | 92.1 (↑1.0) | 0.11 (↓0.17) |

Table 3: Impact of TCM regularization on various DNN models trained with the Recall@k Surrogate loss at a batch size of 4000 as in Patel et al. (2022).

| $Arch.^{dim}$ | R@1 | R@4 | R@16 | OPIS× 1e-3 |
|---|---|---|---|---|
| $ResNet50^{512}$ | 72.2 (↑1.5) | 84.9 (↑1.2) | 92.1 (↑1.0) | 0.11 (↓0.17) |
| $ResNet101^{512}$ | 73.8 (↑1.7) | 85.8 (↑1.1) | 92.6 (↑0.9) | 0.14 (↓0.13) |
| $ViT-S/16^{512}$ | 81.6 (↑0.7) | 90.9 (↑0.5) | 95.6 (↑0.5) | 0.17 (↓0.04) |
| $ViT-B/16^{512}$ | 84.8 (↑0.8) | 92.7 (↑0.6) | 96.5 (↑0.4) | 0.17 (↓0.20) |
| $ViT-L/16^{512}$ | 85.7 (↑0.7) | 93.0 (↑0.7) | 96.6 (↑0.7) | 0.21 (↓0.13) |

Table 4: Time complexities of TCM in comparison to the Recall@k Surrogate loss on the Cars-196 dataset. The ViT-B/16 backbone is utilized with 8× Tesla V100 GPUs and a batch size of 392.

| Method | Complexity | $t_{loss}$ (s) | $t_{backbone}$ (s) | $t_{epoch}$ (s) |
|---|---|---|---|---|
| RS@k | $\mathbb{O}(n^2)$ | 19.9 | 102.6 | 131.3 |
| RS@k + TCM | $\mathbb{O}(n^2)$ | 21.1 | 104.0 | 133.2 |
| Delta | $\mathbb{O}(1)$ | +6.03% | +1.36% | +1.44% |

in the original papers for each base loss. During training, mini-batches are generated by randomly sampling 4 images per class following previous works (Brown et al., 2020; Patel et al., 2022).

## 5.2 ABLATION AND COMPLEXITY ANALYSIS

Unless stated otherwise, all ablation studies are conducted using the iNaturalist-2018 dataset. Owing to space constraints, further ablations can be found in the appendix.

**Effect of TCM margins** We examine the impact of the cosine margins $m^+$, $m^-$ on accuracy and OPIS. As shown in Figure 6, adding TCM consistently enhances threshold consistency compared to the baseline *Smooth-AP* loss across all combinations of margins, with up to 50% of reduction in OPIS. Regarding accuracy, we observe that the negative margin ($m^-$) has a greater influence than the positive margin ($m^+$), which aligns with previous works (Dong et al., 2017; Xuan et al., 2020; Robinson et al., 2020). However, when the negative margin becomes excessively stringent, such as $m^- = 0.25$, the accuracy drops below the baseline. We hypothesize that an overly restrictive negative margin may interfere with the base loss, leading to decreased accuracy. For ImageNet-pretrained backbones, the recommended values for $m^+$ and $m^-$ are around 0.9 and 0.5, respectively.

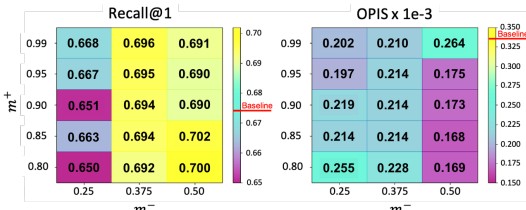

Figure 6: Impact of TCM margins on recall@1 and OPIS for $ResNet50^{512}$ models. The *Smooth-AP* base loss is utilized, yielding baseline results without TCM as recall@1 = 67.4% and OPIS = 3.3e-4.

**Compatibility with various base DML losses** We select the most representative DML losses for each method category, including proxy-based methods (Movshovitz-Attias et al., 2017; Deng et al., 2020) and pairwise-based methods (Brown et al., 2020; Patel et al., 2022). Notably, the *Recall@k surrogate* loss (Patel et al., 2022) represents the SOTA loss for fine-grained image retrieval tasks. We run experiments using these base losses with and without the TCM regularization. As shown in Table 2, there is a consistent improvement in both accuracy (> 1.0% increase in recall@1) and threshold consistency (up to 60.7% in relative reduction) when TCM regularization is applied in conjunction with different high-performing base losses.

**Compatibility with different architectures** We investigate the compatibility of TCM regularization with different backbone architectures including ResNet variants and Vision Transformers. As shown in Table 3, we observe significant improvements in threshold consistency across backbone architectures when TCM is incorporated. On accuracy, ResNet models exhibit more notable improvements in accuracy (> 1.5%) compared to Vision Transformers, which see a < 1.0% boost.

**Time Complexity** In a mini-batch with size $n$, the complexity of TCM is $\mathbb{O}(n^2)$ as it compares every sample with all samples in the mini-batch. For image retrieval benchmarks where the number of training classes $K$ is significantly greater than the batch size $n$, i.e., $K \gg n$, this complexity is comparable to most pair-based losses ($\mathbb{O}(n^2)$) and proxy-based losses ($\mathbb{O}(nK)$). In Table 4, we provide time complexities for the loss computation, the forward and backward passes and the overall

Table 5: Performance of supervised image retrieval after incorporating TCM regularization in recall@k (the higher the better) and OPIS (the lower the better) on 4 image retrieval datasets. The numbers in black represent models trained with $L_{\text{base}} + L_{\text{TCM}}$, while the colored numbers indicate improvement / degradation in absolute magnitude over models trained with $L_{\text{base}}$ alone. For Cars, with the same DINO backbone and ProxyAnchor base loss as in Kim et al. (2023), TCM achieves a R@1 of 91.9, with a 46.3% relative OPIS reduction.

| Benchmark | $\text{Arch}^{dim}$ | $L_{\text{base}} + L_{\text{TCM}}$ | BS | OPIS $_{\times 10^{-3}}\downarrow$ | 10%-OPIS $_{\times 10^{-3}}\downarrow$ | R@1 ↑ | Previous SOTA with ImageNet pretraining |
|---|---|---|---|---|---|---|---|
| **iNaturalist-2018** | ResNet50$^{512}$ | SAP + TCM | 384 | 0.17 ↓0.16(−48.5%) | 1.77 ↓2.83(−61.5%) | 69.1 ↑1.7 | R@1: 83.9 ViT-B/16 (Patel et al., 2022) |
| | | RS@k + TCM | 4000 | 0.11 ↓0.17(−60.7%) | 1.25 ↓2.49(−66.6%) | 72.2 ↑1.5 | |
| | ViT-B/16$^{512}$ | SAP + TCM | 384 | 0.20 ↓0.19(−48.7%) | 2.81 ↓2.40(−46.1%) | 81.2 ↑1.8 | |
| | | RS@k + TCM | 4000 | 0.17 ↓0.20(−54.1%) | 2.03 ↓5.63(−73.5%) | **84.8** ↑0.9 | |
| **Stanford Online Product** | ResNet50$^{512}$ | SAP + TCM | 384 | 0.06 ↓0.11(−64.7%) | 0.52 ↓1.17(−69.2%) | 82.7 ↑2.9 | R@1: 88.0 ViT-B/16 (Patel et al., 2022) |
| | | RS@k + TCM | 4000 | 0.07 ↓0.03(−30.1%) | 0.74 ↓0.12(−14.0%) | 83.3 ↑0.6 | |
| | ViT-B/16$^{512}$ | SAP + TCM | 384 | 0.04 ↓0.01(−25.4%) | 0.33 ↓0.11(−25.0%) | 87.3 ↑0.2 | |
| | | RS@k + HMC | 4000 | 0.04 ↓0.00(−3.7%) | 0.38 ↓0.08(−17.4%) | **88.4** ↑0.4 | |
| **CUB-200-2011** | ResNet50$^{512}$ | SAP + TCM | 384 | 0.11 ↓0.04(−26.7%) | 1.00 ↓0.43(−30.1%) | 80.8 ↑1.0 | R@1: 85.7 ViT-S/16 (Kim et al., 2023) |
| | | RS@k + TCM | 384 | 0.10 ↓0.12(−54.5%) | 0.91 ↓1.04(−53.3%) | 80.0 ↑0.7 | |
| | ViT-B/16$^{512}$ | SAP + TCM | 384 | 0.07 ↓0.14(−66.7%) | 0.58 ↓1.08(−65.1%) | **88.4** ↑0.0 | |
| | | RS@k + TCM | 384 | 0.10 ↓0.34(−77.3%) | 0.91 ↓2.66(−74.5%) | 87.6 ↓0.1 | |
| **Cars-196** | ResNet50$^{512}$ | SAP + TCM | 384 | 0.39 ↓0.06(−13.3%) | 3.33 ↓1.24(−27.1%) | 89.6 ↑2.7 | R@1: **91.3** DINO (Kim et al., 2023) |
| | | RS@k + TCM | 392 | 0.45 ↓0.02(−4.3%) | 2.93 ↓0.65(−18.2%) | 89.7 ↓0.2 | |
| | ViT-B/16$^{512}$ | SAP + TCM | 384 | 0.54 ↓0.66(−55.2%) | 0.83 ↓1.79(−68.3%) | 87.8 ↑0.7 | |
| | | RS@k + TCM | 392 | 0.60 ↓0.37(−38.1%) | 0.98 ↓1.73(−63.8%) | 87.7 ↑0.8 | |

time per epoch. The results suggest that adding TCM regularization results in a negligible ($< 1.5\%$) increment in the overall training time per epoch.

## 5.3 IMAGE RETRIEVAL EXPERIMENT

The results for supervised fine-tuning for image retrieval benchmarks with and without the TCM regularizer are summarized in Table 5. As is shown, our TCM loss is effective in improving threshold consistency (measured by OPIS and $\epsilon$-OPIS, the lower the better), by up to 77.3%, compared to the various baseline losses considered. Meanwhile, adding TCM regularization consistently improves accuracy across almost all benchmarks, base losses and backbone architectures. While we notice a slight decrease in *recall@1* on the two smaller datasets (as marked in red), namely CUB and Cars, these are at the same magnitude as non-significant variations due to random initialization during training. It's worth highlighting that on iNaturalist-2018, arguably the largest public image retrieval benchmark, adding our TCM regularization is shown to out-perform SOTA DML loss, *recall@k surrogate*, reducing the OPIS threshold inconsistency score from $0.37 \times 10^{-3}$ to $0.17 \times 10^{-3}$, while improving the recall@1 accuracy metrics from 83.9% to 84.8%.

## 6 CONCLUSION

In this work, we comprehensively study the issue of threshold inconsistency in deep metric learning. We introduce a novel variance-based metric named Operating-Point-Inconsistency-Score (OPIS) to quantify threshold inconsistency among different classes. Distinct from the One-Threshold-for-All evaluation protocol proposed by Liu et al. (2022), a key advantage of OPIS is its elimination of the need for a separate calibration dataset. As a result, OPIS can be easily utilized alongside existing accuracy metrics, providing an added dimension for evaluating the threshold robustness of trained DML models. With the OPIS metric, we find that achieving high accuracy in a DML model does not necessarily guarantee threshold consistency. To address this issue, we propose the Threshold-Consistent Margin loss (TCM), a simple and versatile regularization technique that can be integrated with any base loss and backbone architecture to improve the model's threshold consistency during training. TCM is designed to enforce more uniform intra-class compactness and inter-class separability across diverse classes in the embedding space. By incorporating TCM, we demonstrate state-of-the-art performance in both threshold consistency and accuracy across various image retrieval benchmarks. We hope that our work serves as a catalyst to encourage more explorations in developing threshold-consistent DML solutions for practical open-world scenarios.

**Limitations of OPIS** The OPIS and $\epsilon$-OPIS metrics necessitate a sufficient number of samples per class to ensure statistical significance, making them unsuitable for few-shot evaluation scenarios.

**Limitations of TCM** Like other inductive deep learning methods, TCM can fail when there's a significant distribution shift between the training and test sets or when strong label noise is present.

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

# A APPENDIX

In the appendix, we offer in-depth theoretical analyses of our proposed OPIS metric (refer to Appendix A.1), conduct additional ablation studies for the TCM regularization (see Appendix A.2), and provide more implementation details (see Appendix A.3).

## A.1 OPERATING-POINT-INCONSISTENCY SCORE

In this section, we provide theoretical analyses for the utility score and the OPIS metric, grounded in Gaussian assumptions. Leveraging these assumptions, we establish upper and lower bounds for OPIS as benchmark values.

### A.1.1 UTILITY SCORE ANALYSIS: A GAUSSIAN MODEL

For a specific class, we assume that its L2 distance distributions between the embeddings of positive sample pairs and between the embeddings of negative sample pairs follow a Gaussian distribution. We represent the L2 distance distribution for positive pairs as $d \sim \mathcal{N}(\mu_{pos}, \sigma_{pos})$ and that for negative pairs as $d \sim \mathcal{N}(\mu_{neg}, \sigma_{neg})$, with the constraint $0 < \mu_{pos} < \mu_{neg} < 2$ for hyperspherical embeddings. Given a distance threshold $d$, we can estimate the fractions of true positive (TP), false negative (FN), false positive (FP) and true negative (TN) as follows:

|  |  | Predicted | |
| :---: | :---: | :---: | :---: |
|  |  | Positive | Negative |
| **Actual** | Positive | $TP = \frac{1+\text{erf}(t_{pos})}{2}$ | $FN = \frac{1-\text{erf}(t_{pos})}{2}$ |
|  | Negative | $FP = \frac{1+\text{erf}(t_{neg})}{2}$ | $TN = \frac{1-\text{erf}(t_{neg})}{2}$ |

Here, $\text{erf}(t)$ represents the Gaussian error function (Andrews, 1998), and $t_{pos} = \frac{d-\mu_{pos}}{\sqrt{2}\sigma_{pos}}$ and $t_{neg} = \frac{d-\mu_{neg}}{\sqrt{2}\sigma_{neg}}$. Thus we can express the accuracy metrics for precision, recall, specificity, and sensitivity as follows:

$$\text{precision} = \frac{1 + \text{erf}(t_{pos})}{2 + \text{erf}(t_{pos}) + \text{erf}(t_{neg})}, \quad \text{recall} = \frac{1}{2}\big(1 + \text{erf}(t_{pos})\big) \tag{7}$$

$$\text{specificity} = \frac{1}{2}\big(1 - \text{erf}(t_{neg})\big), \quad \text{sensitivity} = \frac{1}{2}\big(1 + \text{erf}(t_{pos})\big) \tag{8}$$

Plugging these into the utility score $U_{roc}$ defined by the specificity and sensitivity pair yields:

$$U_{roc}(d) = \frac{(1 + \text{erf}(t_{pos})) \cdot (1 - \text{erf}(t_{neg}))}{2 + \text{erf}(t_{pos}) - \text{erf}(t_{neg})} \tag{9}$$

Similarly, the utility score defined based on the precision and recall pair, denoted as $U_{pr}$ (equivalent to $F_1$ score), can be expressed as the following:

$$U_{pr}(d) = \frac{2 \cdot (1 + \text{erf}(t_{pos}))}{4 + \text{erf}(t_{pos}) + \text{erf}(t_{neg})} \tag{10}$$

Figure 7 gives typical utility score curves. These curves are derived from representative values (indicated in the subtitles) for the mean and standard deviation of the intra-class and inter-class L2 distance distributions of embeddings generated using the Gaussian distribution model. Notably, the left part of the $U_{pr}$ curve and the $U_{roc}$ curve are nearly indistinguishable. This similarity arises because $\text{erf}(t_{neg}) \to -1$ when $t_{neg} \le -2$, leading to $U_{pr}(d) \approx U_{roc}(d) \approx \frac{2}{3+\text{erf}(t_{pos})}$ when $d < 1.0$. Meanwhile, both $U_{roc}$ and $U_{pr}$ utility score curves exhibit a concave shape, indicating a maximum utility attainable at an optimal distance threshold.

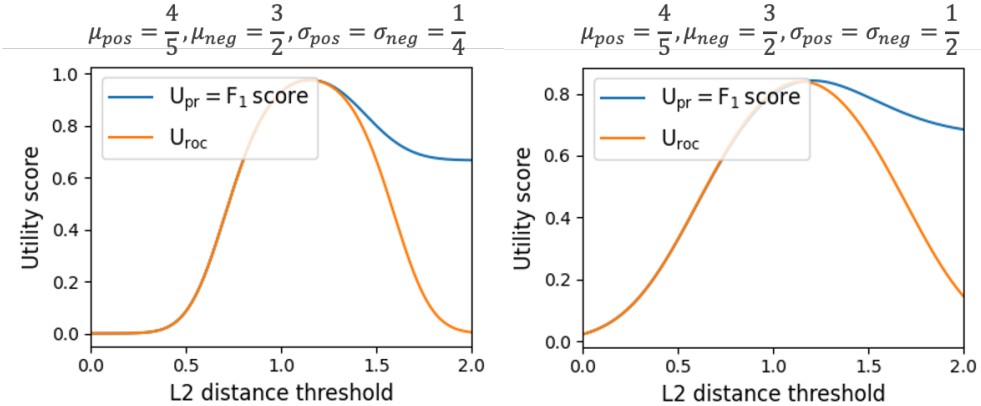

Figure 7: Utility score curves $U_{pr}$ (defined as the harmonic mean of precision and recall) and $U_{roc}$ (defined as the harmonic mean of specificity and sensitivity) plotted as functions of the L2 distance thresholds. The L2 distance distributions of positive and negative embedding pairs are derived from Gaussian distribution models whose means ($\mu$) and standard deviations ($\sigma$) are shown in the subtitles. In general, both $U_{roc}$ and $U_{pr}$ exhibit concavity, indicating a maximum utility is achieved at an optimal distance threshold.

### A.1.2 LOWER AND UPPER BOUNDS OF OPIS

**Lower bound** The OPIS threshold inconsistency metric measures the variance in the utility curves across different test classes. Consequently, its **theoretical lower bound is zero**, a value achieved when utility curves for all test classes are perfectly aligned within the calibration range. However, it's essential to recognize that achieving a zero-variance is an idealistic scenario, unrealistic for real-world datasets. From our evaluations on public image retrieval datasets, **the lower bound of OPIS for prevailing DML losses coupled with ViT backbones is around 1e-5**.

**Absolute upper bound** The upper bound of OPIS depends on the degree of dispersion present in the utility score curves, making it largely dataset-specific. Here, we provide an absolute upper bound. Let's consider an extreme case where a fraction ($= \alpha$) of the test classes are perfectly accurate within the calibration range, i.e., $U_{\text{correct}}(d) = 1, d \in [d_{\min}, d_{\max}]$, while the remainder are entirely inaccurate, i.e., $U_{\text{wrong}}(d) = 0, d \in [d_{\min}, d_{\max}]$. This leads to an average utility of $\bar{U}(d) = \alpha$ and an overall OPIS score of $2\alpha(1 - \alpha)$. This OPIS score has **an absolute upper bound of 0.5**, which is achieved at $\alpha = 50\%$.

**Realistic upper bound** Here we consider a more practical, realistic upper bound for OPIS. Assuming a fraction ($= \alpha$) of the classes have perfect utility score in the calibration range, i.e., $U_{\text{correct}}(d) = 1, d \in [d_{\min}, d_{\max}]$, and the rest are perfectly "random", meaning their pairwise L2 distances are randomly drawn from a uniform distribution with a probability of $p(d = x) = \frac{1}{2}, 0 \leq x \leq 2$. Given these assumptions, the sensitivity and specificity for a random class are described as:

$$\text{sensitivity}(d) = \frac{d}{2}, \quad \text{specificity}(d) = 1 - \frac{d}{2} \tag{11}$$

Incorporating these assumptions into the utility score definition, the utility curves for a "random" class can be expressed as follows[5]:

$$U_{roc}(d) = d(1 - \frac{d}{2}) \tag{12}$$

Thus, the average utility score, represented by $\bar{U}_{\text{roc}}(d)$, can be written as the following:

$$\bar{U}_{\text{roc}}(d) = (1 - \alpha)(1 - \frac{d}{2}) + \alpha \tag{13}$$

---

[5]We focus on $U_{roc}(d)$, the definition used in the main paper, to establish the realistic upper bound of OPIS.

We can then determine the upper bound of OPIS using the equation below:

$$
\begin{aligned}
\text{OPIS} &= \frac{(1-\alpha) \cdot \int_{d_{\min}}^{d_{\max}} ||\bar{\text{U}}(d) - d(1-\frac{d}{2})||^2 \, \mathrm{d}d}{d_{\max} - d_{\min}} + \frac{\alpha \cdot \int_{d_{\min}}^{d_{\max}} ||1 - \bar{\text{U}}(d)||^2 \, \mathrm{d}d}{d_{\max} - d_{\min}} \\
&= \alpha(1-\alpha) \cdot \left(1 + \frac{\int_{d_{\min}}^{d_{\max}} (d-1)^2 \, \mathrm{d}d}{d_{\max} - d_{\min}}\right) \\
&= \frac{\alpha(1-\alpha)}{3} \cdot \left((d_{\max})^2 + (d_{\min})^2 + d_{\max} \cdot d_{\min}\right)
\end{aligned}
\tag{14}
$$

This realistic upper bound $\frac{\alpha(1-\alpha)}{3} \cdot \left((d_{\max})^2 + (d_{\min})^2 + d_{\max} \cdot d_{\min}\right)$ is a function of $\alpha$ and the calibration range. To illustrate, consider a scenario where $\alpha = 10\%$ and $[d_{\min}, d_{\max}] = [0.8, 1.2]$. In this case, the computed OPIS upper bound stands at 9.12e-2.

### A.1.3 SENSITIVITY OF OPIS TO CALIBRATION RANGES

In Figure 8, we show the variation of the OPIS metric across different FAR ranges – each representing a distinct calibration range. Notably, while the absolute values of the OPIS metric might fluctuate across different calibration ranges, the relative rank orderings mostly stay consistent. The only deviation observed is a rank flip between positions 5 and 6 when transitioning from the calibration range of $0.001 < \text{FAR} < 0.05$ to $0.01 < \text{FAR} < 0.1$; meanwhile, ranks 1, 2, 3 and 4 persistently remain stable. It's worth noting that at high FAR, the magnitude of OPIS decreases and variances in relative values will naturally increase when absolute metric values get small. This behavior is attributed to the general tendency for relative ranks to become more unstable as absolute metric values decrease – a phenomenon observed across numerous evaluation metrics including *recall@k* as shown in Brown et al. (2020); Patel et al. (2022); Kim et al. (2020); Movshovitz-Attias et al. (2017); Teh et al. (2020). Given these observations, we consider OPIS to be largely unaffected by the calibration range.

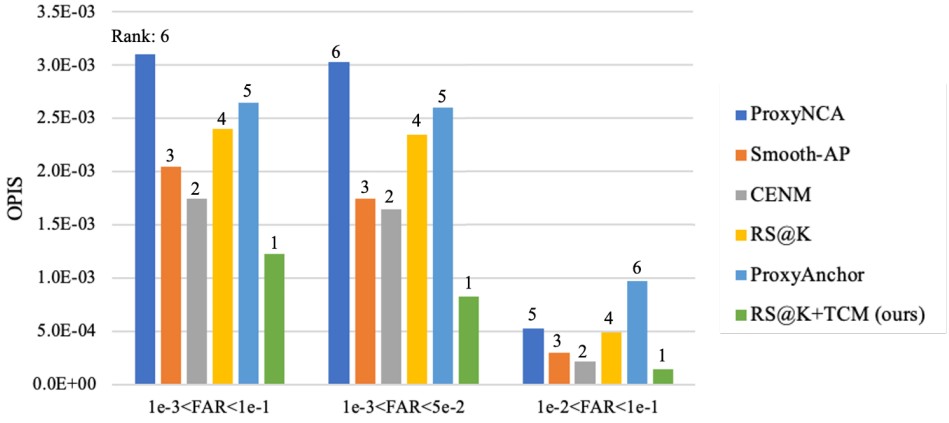

Figure 8: Sensitivity of OPIS to the calibration range: Evaluation of ResNet50$^{512}$ models trained using different losses on the iNaturalist-2018 dataset with a batch size of 384. Different methods are ranked in ascending order according to their OPIS values, with the method exhibiting the lowest OPIS (indicating the best threshold consistency) ranked first.

## A.2 Additional Ablation Studies for TCM

In this section, we present comprehensive ablation studies, including sensitivity analyses, comparison between TCM and other losses, TCM design variations, and an investigation into the compatibility of TCM with self-supervised pretraining. We also give results for the CUB and Cars datasets in an open-world setting.

### A.2.1 Sensitivity of TCM to Random Seeds

Table 6 gives the Recall@k and OPIS evaluation results for ResNet50 models trained using the *Smooth-AP* base loss with the TCM regularization for different random seeds. As demonstrated, the improvements in accuracy and calibration consistency due to the TCM regularization remain stable across varied random seeds, exhibiting a small variation of 0.1 for recall@k and 4e-6 for OPIS.

Table 6: Effect of different random seeds on recall@k and OPIS for models trained on the iNaturalist-2018 dataset, both with and without the TCM regularization, using *Smooth-AP* (SAP) as the base loss. SAP* represents results taken from the original paper (Brown et al., 2020), while **SAP+TCM 1** corresponds to the results we reported in Table 5 of our main paper.

| Method | Recall@1 | @4 | @16 | @32 | OPIS $_{\times 1e\text{-}3}$ |
|---|---|---|---|---|---|
| SAP* | 67.2 | 81.8 | 90.3 | 93.1 | - |
| SAP | 67.4 | 81.9 | 90.4 | 93.1 | 0.330 |
| **SAP+TCM 1** | 69.1 | 82.9 | 91.1 | 93.8 | 0.168 |
| SAP+TCM 2 | 69.0 | 82.9 | 91.0 | 93.7 | 0.172 |
| SAP+TCM 3 | 69.2 | 83.0 | 91.2 | 93.9 | 0.175 |
| **Average** | 69.1± 0.1 | 82.9±0.1 | 91.1 ±0.1 | 93.8±0.1 | 0.172±0.004 |

### A.2.2 Comparison with Other Regularizations

In Table 7, we demonstrate the superiority of our proposed TCM loss over other DML losses, including the *ArcFace* loss (Deng et al., 2019), the *Triplet* loss (Schroff et al., 2015a), and the contrastive loss (Hadsell et al., 2006), when used as a regularizer. It's worth noting that these losses can function as stand-alone DML losses, whereas TCM is specifically designed as a regularizer. As indicated in the table, although adding contrastive loss as the regularizer leads to the best threshold consistency, it also causes some degradation in recall@1. In contrast, our TCM regularization concurrently enhances both accuracy and threshold consistency, resulting in a 1.7% boost in recall@1 and a relative OPIS improvement of 48%.

Table 7: Comparison of our proposed TCM loss with other DML losses (Deng et al., 2019; Schroff et al., 2015a; Hadsell et al., 2006) when employed as a regularizer on the iNaturalist-2018 dataset.

| Method | Arch$^{dim}$ | Batch size | R@1 | @4 | @16 | OPIS $_{\times 1e\text{-}3}$ |
|---|---|---|---|---|---|---|
| SAP | ResNet50$^{512}$ | 384 | 67.4 | 81.9 | 90.4 | 0.330 |
| SAP+ArcFace | ResNet50$^{512}$ | 384 | 62.3 $_{\downarrow 5.1}$ | 77.9 $_{\downarrow 4.0}$ | 87.6 $_{\downarrow 2.8}$ | 0.36 $_{\uparrow 0.03}$ |
| SAP+Triplet | ResNet50$^{512}$ | 384 | 66.2 $_{\downarrow 1.2}$ | 81.3 $_{\downarrow 0.6}$ | 90.1 $_{\downarrow 0.3}$ | 0.35 $_{\uparrow 0.02}$ |
| SAP+Contrastive | ResNet50$^{512}$ | 384 | 67.1 $_{\downarrow 0.3}$ | 82.1 $_{\uparrow 0.2}$ | 90.6 $_{\uparrow 0.2}$ | **0.10** $_{\downarrow 0.23}$ |
| **SAP+TCM (ours)** | ResNet50$^{512}$ | 384 | **69.1** $_{\uparrow 1.7}$ | **82.9** $_{\uparrow 1.0}$ | **91.1** $_{\uparrow 0.7}$ | 0.17 $_{\downarrow 0.16}$ |

We also compare our TCM loss with the calibration loss introduced in ROADMAP (Ramzi et al., 2021), which aims to address the decomposability gap for average precision during training. Their approach, akin to ours, involves the imposition of constraints to regulate pairwise similarity scores. However, a notable difference lies in the denominator's composition: they use the total counts of positive and negative pairs, whereas we employ hard positive and negative counts. An apparent limitation of their design is that in scenarios with a large negative-to-positive ratio, the importance of the negative term diminishes. Such scenarios often arise with large batch sizes (which is the standard of DML), such as the case in Patel et al. (2022) where a batch size of 4000 is employed. This limitation can have adverse effect on the model performance as a body of research (Schroff et al.,

2015b; Oh Song et al., 2016; Wu et al., 2017; Wang et al., 2019) has consistently demonstrated that hard negative pairs generally convey more information than hard positive pairs. As shown in Table 8, when comparing RS@K+TCM and RS@K+ROADMAP at a batch size of 4000, using ROADMAP as a regularization results in a significant reduction in accuracy, aligning with our earlier analysis.

Table 8: Comparison of our proposed TCM loss with ROADMAP at a large batch size.

| Method | $Arch^{dim}$ | Batch size | R@1 | @4 | @16 | OPIS $_{\times 1e\text{-}3}$ |
|---|---|---|---|---|---|---|
| RS@K | ViT-B/16$^{512}$ | 4000 | 83.9 | 92.1 | 96.1 | 0.37 |
| RS@K+ROADMAP | ViT-B/16$^{512}$ | 4000 | 79.4 $_{\downarrow 4.5}$ | 89.2 $_{\downarrow 2.9}$ | 94.3 $_{\downarrow 1.8}$ | 0.24 $_{\downarrow 0.13}$ |
| **RS@K+TCM (ours)** | ViT-B/16$^{512}$ | 4000 | **84.8** $_{\uparrow 0.9}$ | **92.7** $_{\uparrow 0.6}$ | **96.5** $_{\uparrow 0.4}$ | **0.17** $_{\downarrow 0.20}$ |

### A.2.3 COMPARISON OF TCM AND ITS DESIGN VARIANTS

We examine the impact of various components within TCM, namely the indicator function, the positive term, and the negative term, to provide deeper insights into the design of the TCM regularization. Specifically, we consider three design variations, each omitting just one of its three core components: the positive term, the negative term, and the indicator functions. These TCM design alternatives are described as follows:

$$L_{\text{TCM}^-} = \frac{1}{|S^{m^-}|} \sum_{s \in S^-} (s - m^-) \cdot \mathbf{1}_{s \geq m^-} \tag{15}$$

$$L_{\text{TCM}^+} = \frac{1}{|S^{m^+}|} \sum_{s \in S^+} (m^+ - s) \cdot \mathbf{1}_{s \leq m^+} \tag{16}$$

$$L_{\text{TCM}'} = \frac{1}{|S^{m^+}|} \sum_{s \in S^+} (m^+ - s) + \frac{1}{|S^{m^-}|} \sum_{s \in S^-} (s - m^-) \tag{17}$$

where $L_{\text{TCM}^+}$ denotes the variant using only the positive term, $L_{\text{TCM}^-}$ represents the variant with just the negative term, and $L_{\text{TCM}'}$ stands for the variant without the indicator functions.

Table 9: Performance of supervised image retrieval on the iNaturalist-2018 dataset using *recall@k surrogate* loss with TCM regularization and its alternative designs at a batch size of 4000. The results highlight the efficacy of the TCM regularization in comparison to its design alternatives in terms of retaining the accuracy performance while improving OPIS.

| Method | $Arch^{dim}$ | Batch size | R@1 | @4 | @16 | OPIS $_{\times 1e\text{-}3}$ |
|---|---|---|---|---|---|---|
| RS@K | ViT-B/16$^{512}$ | 4000 | 83.9 | 92.1 | 96.1 | 0.37 |
| RS@K+TCM$^+$ | ViT-B/16$^{512}$ | 4000 | 79.0 $_{\downarrow 4.9}$ | 88.9 $_{\downarrow 3.2}$ | 94.2 $_{\downarrow 1.9}$ | **0.06** $_{\downarrow 0.31}$ |
| RS@K+TCM$^-$ | ViT-B/16$^{512}$ | 4000 | 72.5 $_{\downarrow 11.4}$ | 86.0 $_{\downarrow 6.1}$ | 93.0 $_{\downarrow 3.1}$ | 0.14 $_{\downarrow 0.23}$ |
| RS@K+TCM$'$ | ViT-B/16$^{512}$ | 4000 | 78.1 $_{\downarrow 5.8}$ | 88.4 $_{\downarrow 3.7}$ | 93.7 $_{\downarrow 2.4}$ | 0.31 $_{\downarrow 0.06}$ |
| **RS@K+TCM (ours)** | ViT-B/16$^{512}$ | 4000 | **84.8** $_{\uparrow 0.9}$ | **92.7** $_{\uparrow 0.6}$ | **96.5** $_{\uparrow 0.4}$ | 0.17 $_{\downarrow 0.20}$ |

In Figure 9, we give the progression of accuracy across training epochs. The accuracy and OPIS corresponding to the epoch with the peak accuracy are detailed in Table 9. In general, excluding any of the three components of TCM leads to a marked decrease in recall@1 of up to 12%. Moreover, in scenarios where only the positive term is employed or when TCM operates without the indicator function, we observe that the model stops improving its accuracy after the first epoch. When relying solely on TCM's negative term, we observe an initial dip in recall@1, which shows signs of recovery after 20 epochs. Nevertheless, the eventual recall@1, even post-recovery, lags significantly behind that achieved with the full TCM regularization with all three components included. In terms of threshold consistency, our observations suggest that penalizing the TCM positive term, the negative term, or both simultaneously can all improve the OPIS metric. Notably, regularizing solely with the positive TCM term ($L_{\text{TCM}^+}$) yields the best OPIS result. However, omitting the indicator function results in minimal improvements in OPIS, aligning closely with the baseline outcome observed

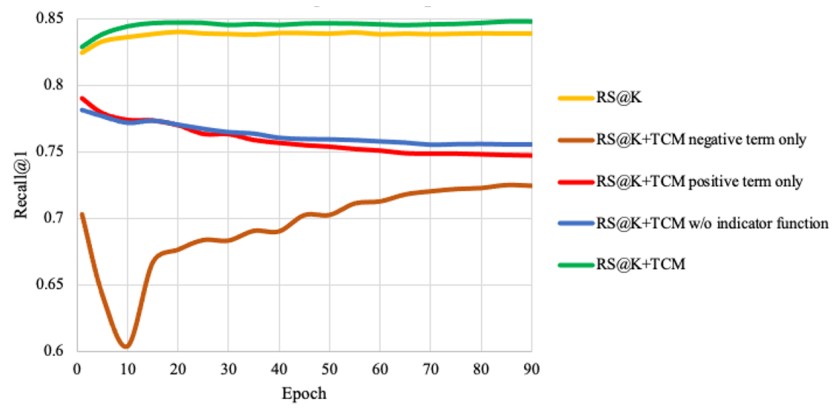

Figure 9: Evolution of recall@1 across training epochs for ViT-B/16 backbones trained on the iNaturalist-2018 dataset at a batch size of 4000. The Recall@k surrogate loss is used as the base loss along with the TCM regularization and its design alternatives.

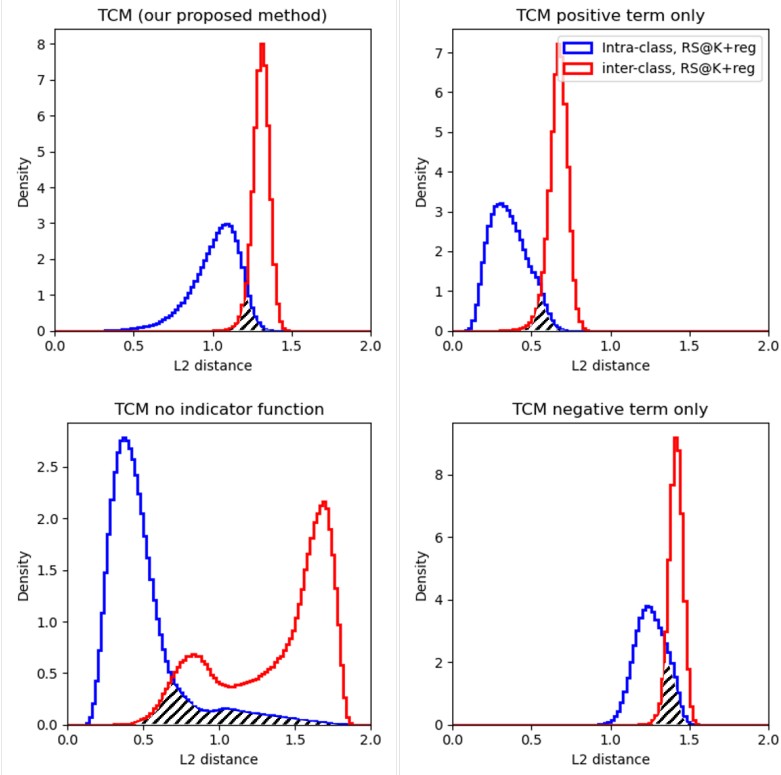

Figure 10: Distributions of L2 distances for intra-class and inter-class test embeddings of ViT-B/16 models trained on the iNaturalist-2018 dataset using the *recall@k surrogate* loss with TCM regularization and its variants. It shows that excluding any of the TCM loss components introduces irregularities in the metric space, therefore diminishing accuracy.

without any regularization. This showcases the importance of the indicator function as an effective quantile estimation mechanism to filter out the hard sample pairs.

We also visualize the intra-class and inter-class L2 distance distributions in Figure 10 for models trained with TCM regularization and the other four alternative designs. The results reveal that a regularizer focusing solely on penalizing hard negative pairs (RS@K+TCM$^-$) causes the inter-class embeddings to become overly dispersed, compromising the desired intra-class compactness. Con-

versely, when only hard positive pairs are penalized (RS@K+TCM$^+$), it results in the collapse of intra-class embeddings, leaving the inter-class distribution unchecked. On the other hand, in the absence of the indicator function, the inter-class distribution manifests as multi-modal with dual peaks with significantly increased overlap between the inter-class and intra-class L2 distance distributions. This phenomenon may be attributed to the failure to distinguish hard negative pairs from the easy ones. To conclude, to maintain accuracy on par with the baseline (without regularization), all three components of TCM are indispensable.

### A.2.4   COMPATIBILITY OF TCM WITH SELF-SUPERVISED PRETRAINING

We examine the compatibility of our proposed TCM regularization with self-supervised pretraining methods, including CLIP (Radford et al., 2021), DINOv2 (Oquab et al., 2023), and Unicom (An et al., 2023). In alignment with the approach in Unicom An et al. (2023), we adopt the ArcFace loss (Deng et al., 2019) for this supervised fine-tuning. The results are presented in Table 10.

Table 10: Performance of supervised image retrieval on the Stanford Online Product dataset with different pretraining methods including CLIP (Radford et al., 2021), DINOv2 (Oquab et al., 2023) and Unicom (An et al., 2023). The pretraining dataset for each backbone model is specified in the text next to each method. Following Unicom (An et al., 2023), we use the ArcFace loss (Deng et al., 2019) for supervised fine-tuning. Same as Table 5 in the the main paper, the numbers displayed in black represent models trained with $L_{\text{base}} + L_{\text{TCM}}$, while the colored numbers indicate the improvement / degradation in absolute magnitude over models trained with the base loss alone.

| Pretraining | Pretraining data | Arch$^{dim}$ | R@1 ↑ | OPIS $\times 10^{-3}$ ↓ |
|---|---|---|---|---|
| CLIP | Private400M | ViT-B/16$^{512}$ | 88.4 ↑4.50 (+5.3%) | 0.94 ↓0.06 (−6.2%) |
| DINOv2 | LVD-142M | ViT-B/14$^{768}$ | 84.2 ↑0.20 (+0.2%) | 0.84 ↓0.09 (−9.7%) |
| Unicom | LAION-400M | ViT-B/16$^{768}$ | 89.1 ↑0.30 (+0.3%) | 0.82 ↓0.10 (−10.9%) |

As shown in the table, incorporating TCM regularization in models initialized through various self-supervised pretraining methods consistently delivers improvements in the OPIS metrics, observing gains of up to 10%. However, the relative improvement in OPIS doesn't quite reach the levels achieved with supervised pretraining as reported in the main paper. This discrepancy is likely caused by the distinctive learning objectives of self-supervised pretraining and its use of completely different pretraining datasets compared to ImageNet. As for accuracy, adding TCM is shown to significantly boost recall@1 by up to 4.5% when paired with CLIP pretraining. Yet, for approaches such as DINOv2 and Unicom, the enhancement in recall@1 is marginal.

### A.2.5   OPEN-WORLD SETTING RESULTS FOR CUB AND CARS

In the main paper, we utilize shared train-test classes in the closed-set setting for the CUB and Cars datasets to ensure fair comparisons with previous works. However, in this section, we present results for these two datasets under the open-world setting, where training and testing classes are disjoint, to align with our central focus on open-world scenarios. Specifically, we partition the datasets based on the class indices: for CUB, classes 1-100 serve as the training set while the rest constitute the test set; similarly, for Cars, classes 1-98 are used for training, with the remaining classes reserved for testing. In Table 11, we present evaluation results for the CUB and Cars datasets under the open-world setting. The results demonstrate that integrating TCM leads to a notable relative improvement of 18% in OPIS for the Cars-196 dataset and a substantial 45% enhancement for the CUB dataset, while preserving and improving the accuracy compared to the baseline without regularization.

Table 11: Evaluation results of incorporating TCM regularization on recall@k and OPIS for the CUB and Cars-196 datasets in the open-world setting.

| Dataset | Arch$^{dim}$ | R@1 ↑ | OPIS $\times 10^{-3}$ ↓ | 10%-OPIS $\times 10^{-3}$ ↓ |
|---|---|---|---|---|
| CUB | ViT-B/16$^{512}$ | 86.5 ↑0.0 | 0.92 ↓0.76 (−45.2%) | 10.43 ↓9.04 (−46.4%) |
| Cars-196 | ViT-B/16$^{512}$ | 90.2 ↑0.7 | 0.63 ↓0.14 (−18.2%) | 8.07 ↓2.33 (−22.4%) |

## A.3 IMPLEMENTATION DETAILS

### A.3.1 CONNECTIONS BETWEEN CALIBRATION RANGE AND COSINE MARGINS

We clarify the connection between the calibration range and the cosine margins introduced in TCM. First, **calibration range is determined by the FAR and FRR requirement specified by the use-case.** For a well-trained embedding model, optimal performance is typically achieved within an intermediate range of distance thresholds. Thus, the left bound of the calibration range is determined by the maximum FRR, as defined by the use-case. On the flip side, the right bound is determined by the maximum FPR set by the same use-case. The margins $m^+$ and $m^-$ of the TCM regularization are defined in the cosine space. Given hyperspherical embeddings, there is a bijection between the cosine similarity and the L2 distance. In the absence of a distribution shift between the training and testing datasets, the cosine margins can be adeptly pinpointed as $m^+ = \frac{\sqrt{2-d_{\min}^2}}{2}$ and $m^- = \frac{\sqrt{2-d_{\max}^2}}{2}$ to align the representation structures to the desired calibration range. However, while the calibration range can serve as a useful reference, due to the inevitable distribution shifts between training and testing data, the selection of the margin constraints requires some hyper-parameter tuning and cannot be directly estimated from the calibration range.

### A.3.2 ELABORATION ON FIG.5(B) OF THE MAIN PAPER

In Fig.5(b) of the main paper, $\theta_1$ and $\theta_2$ represent the angular arc lengths of the red and blue classes, respectively. Applying TCM regularization promotes intra-class compactness, targeting a condition where $cos(\theta) = m^+$, with $\theta$ representing the angular span of the given class in the embedding. Thus training with TCM regularization would encourage $\theta_1 \leftarrow \arccos(m^+)$ and $\theta_2 \leftarrow \arccos(m^+)$.

### A.3.3 IMPLEMENTATION DETAILS FOR OPIS METRIC AND TCM REGULARIZATION

In the following, we provide a brief description of our implementation for both the OPIS metric and the TCM regularization in pseudo-code format.

---

**Algorithm 1** Computation for OPIS metric

---

**Require:** Pairwise evaluation protocol for test embeddings with a total of $T$ classes, calibration range $[d_{\min}, d_{\max}]$ with a grid number $N$.

1: Initialize OPIS $\leftarrow 0$
2: **for** $j = 1$ to $N$ **do**                 ▷ Iterate over calibration grid
3:     $d \leftarrow d_{\min} + \frac{d_{\max}-d_{\min}}{n} \cdot j$
4:     **for** $i$ in $T$ **do**                 ▷ Iterate over test classes
5:         Gather all L2 distances for class $i$
6:         Compute specificity ($\phi$) and sensitivity ($\psi$) at $d$
7:         $U_i(d) \leftarrow \frac{2 \cdot \phi(d) \cdot \psi(d)}{\phi(d)+\psi(d)}$
8:     Compute average utility score $\bar{U}(d)$ across all classes
9:     OPIS $\leftarrow$ OPIS $+ \frac{\sum_{i=1}^{T} ||U_i(d)-\bar{U}(d)||_2^2}{T \cdot (d_{\max}-d_{\min})}$
    **return** OPIS

---

In **Algorithm 1**, the number of grids in the calibration range, denoted as $N$, should be selected in accordance with the maximum sharpness observed in the utility-distance curves within the calibration range. It is recommended to maintain $N \geq 10$. The calibration range itself is determined by the desired False Acceptance Rate (FAR) range of the entire dataset, as specified by the user case. Additionally, it is bound by the least achievable FAR of the dataset. In **Algorithm 2**, $|S^{m^+}|$ represents the number of hard positive pairs in $S^+$ with similarity below $m^+$ and $|S^{m^-}|$ represents the number of hard negative pairs in $S^-$ with similarity above $m^-$.

---

**Algorithm 2** Training with TCM regularization

---

**Require:** $m^+, m^-, \lambda^+, \lambda^-$, base loss $L_{\text{base}}$.

1: **for** number of epochs **do**
2:      **for** $k$ steps **do**                                                ▷ In each training iteration
3:          $L_{\text{TCM}} \leftarrow 0$
4:          Sample mini-batch data $\{(x_1, y_1), ..., (x_m, y_m)\}$
5:          Extract embeddings (denoted by $f$) for each $x$
6:          Compute cosine similarities for the entire mini-batch data. Let $S^+$ be the set of positive pair similarities and $S^-$ be the set of negative pair similarities
7:          Compute $|S^{m^+}|$ and $|S^{m^-}|$
8:          **for** $s$ in $S^+$ **do**
9:              $L_{\text{TCM}} \leftarrow L_{\text{TCM}} + \lambda^+ \cdot \frac{(m^+ - s) \cdot \mathbf{1}_{s \leq m^+}}{|S^{m^+}|}$             ▷ Sum over all hard positive pairs
10:        **for** $s$ in $S^-$ **do**
11:            $L_{\text{TCM}} \leftarrow L_{\text{TCM}} + \lambda^- \cdot \frac{(s - m^-) \cdot \mathbf{1}_{s \geq m^-}}{|S^{m^-}|}$            ▷ Sum over all hard negative pairs
12:          $L_{\text{overall}} \leftarrow L_{\text{base}} + L_{\text{TCM}}$
13:          Update model using $L_{\text{overall}}$

---

### A.3.4    DETAILS ON THE BASE DML LOSSES EMPLOYED

***Smooth-AP*** *Smooth-AP* (Brown et al., 2020) is a pair-based DML loss that optimizes a smoothed approximation of average precision (AP) using a sigmoid function. It is expressed as:

$$L_{\text{SAP}} = \frac{1}{n} \sum_{i=1}^{n} (1 - \frac{1}{|S_i^+|} \sum_{j \in S_i^+} \frac{1 + \sum_{k \in S_i^+} \mathcal{G}_{jk,i}}{1 + \sum_{k \in S_i} \mathcal{G}_{jk,i}}) \tag{18}$$

Here, $\mathcal{G}_{jk,i} = (1 + e^{\lambda(s_{ij} - s_{ik})})^{-1}$ is a softmax function incorporating a temperature parameter, $\lambda$, to rank similarity scores of mini-batch samples against an anchor sample $x_i$. $S_i$ represents the set of cosine similarity scores for the entire mini-batch against $x_i$, while $S_i^+$ denotes the subset of $S_i$ for the positive samples against $x_i$,

**Recall@k Surrogate** The *Recall@k Surrogate* loss mirrors Smooth-AP's approach but approximates the Recall@k metric rather than the average precision metric. Its detailed formulation is available in Patel et al. (2022).

**ArcFace** *ArcFace* (Deng et al., 2019) introduces additive margin penalties based on the angle, $\theta$, between mini-batch samples and prototype representations. It aims to contrast samples against class prototypes, and is formulated as:

$$L_{\text{ArcFace}} = \frac{1}{n} \sum_{i=1}^{n} \log(1 + \sum_{j=1, j \neq i}^{n} e^{\lambda(\mathfrak{s}_{ij} - \mathfrak{s}_{ii} + \delta)}) \tag{19}$$

where $\mathfrak{s}_{ij}$ is the cosine similarity with inter-class margin penalty between the representation of an image with class $i$ and the prototypical representation for another image whose class is $j$, with $m_1, m_2$ denoting different angular margins:

$$\mathfrak{s}_{ij} = \begin{cases} \cos(m_1 \theta_{ij} + m_2) & \text{for } i = j \\ \cos \theta_{ij}, & \text{for } i \neq j \end{cases} \tag{20}$$

