# OpenReview forum: "Threshold-Consistent Margin Loss for Open-World Deep Metric Learning"
_ICLR.cc/2024/Conference — ICLR 2024 poster_

### Official Review · Reviewer_3dmW · 2023-10-29

**Soundness:** 2 fair
**Presentation:** 4 excellent
**Contribution:** 3 good
**Rating:** 5
**Confidence:** 4

**Summary:**

This paper addresses the issue of inconsistency in threshold determination for negative samples in threshold-based image retrieval. The authors propose a new metric called Operating-Point-Inconsistency-Score (OPIS) to measure inconsistency and introduce the Threshold-Consistent Margin (TCM) loss as a regularization technique to enhance consistency. The key contributions include identifying the problem with existing method, introducing an intuitive evaluation metric and regularization approach, and demonstrating improved threshold consistency without sacrificing accuracy in large-scale experiments.

**Strengths:**

- The paper is well-written, making it easy to understand while offering comprehensive comparisons with current methods.

- It clearly highlights issues in existing models and presents an intuitive metric and regularization technique to tackle them.

- The research goes a step further by demonstrating not just improved threshold consistency but also better performance in several instances.

**Weaknesses:**

- The biggest weakness in the paper is the lack of experiments related to face verification, where threshold importance is evident. While image retrieval mostly uses metrics like mAP or Recall@k, face verification relies heavily on thresholds and uses metrics like TAR@FAR. The introduced method appears more suited for face verification than image retrieval.

- The paper suggests that high accuracy doesn't always mean high threshold consistency. However, in face verification tasks, consistency in threshold often translates to high accuracy. This amplifies the sense that the paper might be focusing on an unrelated task.

- The paper mentions related works like Liu et al. (2022) and OneFace, but experiments comparing the proposed method to these in the realm of face recognition are missing. Such comparisons are necessary to understand the proposed method's improvements in threshold consistency.

- The paper needs to update state-of-the-art results on the CUB and Cars-196 datasets [1].
[1] Kim et al., HIER: Metric Learning Beyond Class Labels via Hierarchical Regularization, CVPR 23

**Questions:**

Figure 3 shows ProxyAnchor (ResNet50) with a low threshold consistency. It would be beneficial to compare the improvement in R@K and OPIS when using the proposed method. Ideally, the proposed method should show a significant OPIS improvement compared to others.

---

> ### Author Response · Authors · 2023-11-19
> **Response to reviewer 3dmW (Part-1 out of 2)**
>
> We thank you for the insightful comments. We are glad that you found our paper to be well-written and easy to understand. We have taken your feedback into account and incorporated updates, including the latest state-of-the-art results on the CUB and Cars-196 datasets, into our manuscript. We have carefully considered and addressed each of your concerns point-by-point, as detailed below.
>
> **Q1/Q3 Lack of experiments related to face verification**
>
> **A1** We sincerely appreciate the reviewer's suggestion and input. However, we would like to point out that the focus of our work is to introduce the notion of threshold consistency for image retrieval in the general image domain and bring awareness of the threshold inconsistency phenomenon in deep metric learning. We want to emphasize the crucial role that the threshold plays in real-world image retrieval systems, where a threshold-based retrieval criterion is often preferred over a top-k approach due to its ability to identify negative queries that do not have matches in the gallery. However, the issue of threshold inconsistency in DML can often complicate the process of selecting thresholds when deploying large-scale commercial image retrieval systems in practice. To address this practical challenge, our paper presents two key contributions: the OPIS metric and the TCM regularization loss, both designed to mitigate the threshold inconsistency problem in deep metric learning.
>
> Regarding the application of our work to face verification, we acknowledge the relevance of threshold consistency in this context. However, due to the potential significant legal risks with using technologies that may generate or process biometric information, we are not able to use datasets where we cannot ensure that our use the datasets complies with applicable law, such as the [Illinois Biometric Information Privacy Act](https://www.ilga.gov/legislation/ilcs/ilcs3.asp?ActID=3004&ChapterID=57) (BIPA). For example, prior to using a dataset for such purposes, we must determine where the data subjects reside and confirm that the data subjects received the required notice and provided the required consent for us to use their data for our proposed purpose. Because we cannot ensure that the proposed use of the face datasets for face verification experiments complies with applicable laws, we are unable to use these datasets as you request. In addition, noteworthy organizations like National Institute of Standards and Technology (NIST) have discontinued the distribution of all three IJB datasets (IJB-A, IJB-B, and IJB-C) required for standard face recognition evaluation in the literature (refer to [link](https://www.nist.gov/programs-projects/face-challenges)). We kindly request suggestions from the reviewer on alternative experiments that we can undertake to address your concern without compromising privacy or violating applicable laws. Your insights and recommendations would be greatly appreciated.
>
>
> **Q2 Relationship between threshold consistency and accuracy in face recognition**
>
> **A2** To our best knowledge, high threshold consistency does not necessarily translate to high accuracy in face recognition. Through an investigation into face literature, it appears that OneFace [1] is the only work that delves into the discussion of threshold consistency across different demographic groups in face recognition. The experiment results in [1] indicate that high threshold consistency does not always translate to high accuracy. CurricularFace [2] exhibits better threshold consistency than GroupFace [3], but GroupFace achieves higher accuracy than CurricularFace on IJB-B and IJB-C in TAR@FAR=1e-4.
>
>
> **Q4 The state-of-the-art results on the CUB and Cars-196 datasets**
>
> **A4** Thank you for pointing out the state-of-the-art methods. We appreciate your suggestion to ensure the accurate representation of our paper. We have promptly updated the numbers in Table 5 of the main paper to reflect the latest progress. For a fair comparison, we also conducted experiments using the same backbone (DINO) as HIER [4] on the Cars dataset. As indicated in the table, TCM outperforms HIER in both OPIS and accuracy, all while reduces training time. We acknowledge and regret any inadvertent oversight in not being initially aware of the SOTA method and we appreciate its identification being brought to our attention.
>
> Table 1. The performance of HIER and our proposed method on Cars in Recall@1 and OPIS.
>
> |  | Backbone-embedding dimension	 |Recall@1	| OPIS x 1e-3	  | Time for Loss Computation (s / iteration)|
> |:-----------------:|:---------:|:---:|:-------------:|:-------------------------------------------:|
> | ProxyAnchor+HIER| DINO-384|91.7 *| 0.55|0.41|
> | ProxyAnchor+TCM| DINO-384 |**91.9**|**0.41**|**0.22**|
>
> *We use the released codebase and use the same hyper-parameter described in [4], and our reproduced result is better than the Recall@1 of 91.3 reported in [4].

---

> ### Author Response · Authors · 2023-11-19
> **Response to reviewer 3dmW (Part-2 out of 2)**
>
> **Q5 Apply TCM on top of ProxyAnchor(ResNet50)**
>
> **A5** We apply TCM on top of ProxyAnchor as required. As illustrated in the table and hypothesized by the reviewer, our proposed method demonstrates a significant performance improvement over the base loss in both Recall@1 and OPIS on the iNaturalist dataset.
>
> Table 2. The performance of applying TCM on top of ProxyAnchor on iNaturalist-2018 in Recall@1 and OPIS.
>
> |  |Backbone-embedding dimension | Recall@1 |OPIS x 1e-3|Relative improvement in OPIS after incorprating TCM as a regularizer|
> |:-------------------:|:---------------:|:-------------:|:--------------:|:----------------------------------------------------------------------:|
> | ProxyAnchor + TCM  | ResNet50-512 | 65.7 (↑6.6) | 0.22 (↓0.76) | 77.60%  |
>
>
> References for Response to reviewer 3dmW:
>
> [1] Liu, Jiaheng, et al. "OneFace: one threshold for all." *European Conference on Computer Vision*. Cham: Springer Nature Switzerland, 2022.
>
> [2] Huang, Yuge, et al. "Curricularface: adaptive curriculum learning loss for deep face recognition." *proceedings of the IEEE/CVF conference on computer vision and pattern recognition*. 2020.
>
> [3] Kim, Yonghyun, et al. "Groupface: Learning latent groups and constructing group-based representations for face recognition." *Proceedings of the IEEE/CVF Conference on Computer Vision and Pattern Recognition*. 2020.
>
> [4] Kim, Sungyeon, Boseung Jeong, and Suha Kwak. "HIER: Metric Learning Beyond Class Labels via Hierarchical Regularization." *Proceedings of the IEEE/CVF Conference on Computer Vision and Pattern Recognition*. 2023.

---

> ### Author Response · Authors · 2023-11-21
> **Looking forward for further discussions**
>
> We would like to extend our sincere gratitude for your dedicated review of our paper. To address your concerns and questions, we have provided additional explanations and experiment results, including a comparison between TCM and HIER, where our proposed TCM method demonstrates competitive performance. Additionally, we have made necessary changes to the manuscript by updating the correct state-of-the-art numbers for Cars and CUB, along with the corresponding citations (HIER, CVPR23). As we are approaching the end of the discussion period, please let us know if you have any additional questions or require further clarifications. Once again, we thank you for your valuable comments and insights.

---

> ### Author Response · Authors · 2023-11-23
> **Additional response for Q3 on applicability of TCM in face domain**
>
> We would like to express our gratitude once again for your valuable reviews. We are committed to fully addressing all of your concerns. To avoid potential legal risks associated with using real human face datasets, we have opted to utilize a synthetic face dataset, DigiFace [5], to demonstrate the effectiveness of our TCM method in comparison with OneFace in the realm of face verification. Same as the open-world setting described in the main paper, we randomly split 90% of person identities for training, reserving the remaining 10% of identities for testing. The training and testing identities (classes) are completely disjoint to each other. The results in the table below gives a comparison ArcFace [6], OneFace [1] and our proposed TCM method. Notably, as OneFace does not open-source their code, we followed their paper and implemented based on the description in the paper. As shown in the table, our method outperforms both ArcFace and OneFace in terms of threshold consistency (measured by OPIS, the lower the better), while achieving the highest accuracy (measured by TAR@FAR=1e-4, the higher the better) among the three. We hope this experiment adequately addresses your concerns about experiments in face verification. Please do not hesitate to reach out if you have any additional questions or concerns.
>
> Table 3. The performance of ArcFace, OneFace and our proposed method on DigiFace in TAR@FAR=1e-4 and OPIS.
>
> |        	        |    Backbone-embedding dimension	    | TAR@FAR=1e-4 (%)	  | OPIS x 1e-2	  |
> |:---------------:|:---------------:|:-----------------:|:-------------:|
> |    ArcFace	     | ResNet100-128	  |       98.20	       |     1.42	     |
> |    OneFace	     | ResNet100-128	  |      99.03	       |     1.41	     |
> | ArcFace + TCM	  | ResNet100-128	  |      99.54	       |     1.28	     |
>
>
> References for response to reviewer 3dmW:
>
> [5] Bae, Gwangbin, et al. "Digiface-1m: 1 million digital face images for face recognition." *Proceedings of the IEEE/CVF Winter Conference on Applications of Computer Vision*. 2023.
>
> [6] Deng, Jiankang, et al. "Arcface: Additive angular margin loss for deep face recognition." *Proceedings of the IEEE/CVF conference on computer vision and pattern recognition*. 2019.

---

### Official Review · Reviewer_52BZ · 2023-10-29

**Soundness:** 3 good
**Presentation:** 3 good
**Contribution:** 3 good
**Rating:** 8
**Confidence:** 5

**Summary:**

The paper introduces and addresses the threshold inconsistency problem in Deep Metric Learning (DML). To tackle this issue, the authors present the Operating-Point-Inconsistency-Score (OPIS) metric, which is based on the variance of utility score derived from the F-score. Additionally, they propose the Threshold-Consistent Margin (TCM) loss, which selectively penalizes hard sample pairs. The experimental results on various deep metric learning benchmarks validate the efficacy of their proposed method.

**Strengths:**

1.The paper effectively identifies and defines the threshold inconsistency problem within the context of Deep Metric Learning (DML).

2.To address this issue, the authors introduce a novel loss function, the Threshold-Consistent Margin (TCM) loss.

3.Their proposed method is rigorously evaluated through comprehensive experiments.

**Weaknesses:**

1. The use of the term "large-scale" in this paper may be misleading as the experiment datasets do not contain a sufficiently large number of samples to be accurately characterized as "large-scale." Typically, datasets with more than 10 million or 1 billion samples could be considered as large-scale.

2. The threshold inconsistency problem, as described in this paper, is also referred to as the generalization problem and has been previously discussed in the deep metric learning (DML) literature [r1, r2]. In reference [r1], the authors proposed the adoption of a metric variance constraint (MVC) to enhance generalization ability, which is essentially a variance-based metric. Reference [r2] provided an in-depth discussion of the generalization problem in DML. It would be beneficial for this paper to incorporate discussions and comparisons with these existing works in the context of addressing the threshold inconsistency problem.

[r1] Kan, Shichao, et al. "Contrastive Bayesian Analysis for Deep Metric Learning." IEEE Transactions on Pattern Analysis and Machine Intelligence (2022).

[r2] Karsten Roth, Timo Milbich, Samarth Sinha, Prateek Gupta, Björn Ommer, and Joseph Paul Cohen. Revisiting training strategies and generalization performance in deep metric learning. In Proceedings of the 37th International Conference on Machine Learning, ICML 2020, 13-18 July 2020, Virtual Event, volume 119 of Proceedings of Machine Learning Research, pages 8242–8252, 2020.

**Questions:**

See the weaknesses.

---

> ### Author Response · Authors · 2023-11-19
> **Response to reviewer 52BZ (Part-1 out of 2)**
>
> We thank you for the encouraging review and constructive comments to enhance our paper. It is gratifying to hear you find that we “effectively identify and define the threshold inconsistency problem in DML”, “introduce a novel loss function” and “ rigorously evaluate our method through comprehensive experiments”. We response to your comments below.
>
> **Q1 The use of the term "large-scale" in this paper may be misleading**
>
> **A1** Thanks for the valuable comments. In our experiments, we evaluated our models on standard image retrieval benchmarks, aligning with practices in mainstream deep metric learning studies [1, 2, 3, 4]. We follow previous works [5, 6, 7] which used the word "large-scale" to describe the iNaturalist-2018 dataset, which is much bigger than traditional metric learning datasets such as CUB, Cars, and etc. However, we acknowledge that the scale of iNaturalist-2018 is limited compared to many recently-released datasets such as LAION-400M and LAION-5B, although these datasets are typically not used for metric learning. To enhance clarity and better reflect the scale of our experiments, we have incorporated the suggested revisions into the manuscript by taking out the term "large-scale experiments".

---

> ### Author Response · Authors · 2023-11-19
> **Response to reviewer 52BZ (Part-2 out of 2)**
>
> **Q2 Discussions and comparisons about [1, 2]**
>
> **A2** We genuinely appreciate the reviewer for sharing the two referenced works with us. Through our examination, we have indeed identified noteworthy parallels between the generalization problem discussed in [1, 2] and our open-world setting. Particularly, one of conclusions in [2], stating that excessive feature compression actually hurts DML generalization, aligns with our observations in the ablation study for margin, as shown in Fig 6 of the paper, where overly restrictive margins lead to decreased accuracy. We have promptly incorporated references to these two works into our revised paper to further enrich the context and related research in the field. However, it's worth noting that these two works primarily focus on enhancing generalization to improve accuracy, particularly in scenarios characterized by train-test distribution or concept shifts. In contrast, our work specifically targets the crucial aspect of threshold consistency. Our proposed evaluation criteria, OPIS and epsilon-OPIS, are tailored to measure threshold consistency within the calibration range across different test distributions and classes. They are designed as orthogonal metrics to accuracy.
>
> Additionally, comparing TCM to the Margin Variance Control (MVC) proposed in CBML[1], we find commonalities between the two techniques. However, we would like to point out the three distinctions between TCM and MVC that makes TCM more suited for encouraging threshold consistency:
>
> 1. TCM employs two cosine margins to simultaneously regulate positive and negative pairs, whereas MVC is exclusively applied to negative pairs. In Appendix A.2.3, we demonstrate that relying solely on the negative term lags considerably behind the full TCM regularization in terms of Recall@1 (72.5 for negative term only vs. 84.4 for both positive and negative terms).
> 2. TCM applies penalties to hard sample pairs, while MVC minimizes variance for all negative pairs. In Appendix A.2.3, we consider a TCM variant without hard mining (referred to as TCM’ in Appendix A.2.3) by removing the indicator function in Eq. (5), where the regularization is applied to all positive and negative pairs. However, as shown in Appendix A.2.3, the omission of the indicator function yields a significant degradation in Recall@1 (74.1 for RS@K + TCM without indicator function vs. 84.8 for RS@K + TCM with indicator function, the higher the better), while producing a worse OPIS result (0.31 x 1e-3 for RS@K + TCM without indicator function vs 0.17 x 1e-3 for RS@K + TCM with indicator function, the lower the better). This result underscores the critical importance of hard sample mining.
> 3. TCM utilizes a global margin, whereas MVC calculates sample-specific margin, namely, target value for the similarity hyperplane as defined in [1].
>
>
> References for Response to reviewer 52BZ:
>
> [1] Kan, Shichao, et al. "Contrastive Bayesian Analysis for Deep Metric Learning." IEEE Transactions on Pattern Analysis and Machine Intelligence (2022).
>
> [2] Karsten Roth, Timo Milbich, Samarth Sinha, Prateek Gupta, Björn Ommer, and Joseph Paul Cohen. Revisiting training strategies and generalization performance in deep metric learning. In Proceedings of the 37th International Conference on Machine Learning, ICML 2020, 13-18 July 2020, Virtual Event, volume 119 of Proceedings of Machine Learning Research, pages 8242–8252, 2020.
>
> [3] Brown, Andrew, et al. "Smooth-ap: Smoothing the path towards large-scale image retrieval." *European Conference on Computer Vision*. Cham: Springer International Publishing, 2020.
>
> [4] Patel, Yash, Giorgos Tolias, and Jiří Matas. "Recall@ k surrogate loss with large batches and similarity mixup." *Proceedings of the IEEE/CVF Conference on Computer Vision and Pattern Recognition*. 2022.
>
> [5] Cui, Yin, et al. "Class-balanced loss based on effective number of samples." *Proceedings of the IEEE/CVF conference on computer vision and pattern recognition*. 2019.
>
> [6] Zhou, Boyan, et al. "Bbn: Bilateral-branch network with cumulative learning for long-tailed visual recognition." *Proceedings of the IEEE/CVF conference on computer vision and pattern recognition*. 2020.
>
> [7] Zhong, Zhisheng, et al. "Improving calibration for long-tailed recognition." *Proceedings of the IEEE/CVF conference on computer vision and pattern recognition*. 2021.

---

> ### Author Response · Authors · 2023-11-21
> **Looking forward for further discussions**
>
> We extend our sincere appreciation for your valuable feedback. We have taken out the word “large-scale” and incorporated discussions regarding the two suggested works into the updated manuscript. Please kindly provide feedback on whether these updates have addressed your inquiries. We genuinely thank you once again for dedicating your time and expertise to review our work.

---

> > ### Comment · Reviewer_52BZ · 2023-11-21
> > **Thank you for the response!**
> >
> > Thank you for the thoughtful response to my review and others. I will keep my current rating and hope to see this paper accepted.

---

### Official Review · Reviewer_4pGE · 2023-10-31

**Soundness:** 3 good
**Presentation:** 3 good
**Contribution:** 3 good
**Rating:** 6
**Confidence:** 4

**Summary:**

This paper addresses the problem of threshold inconsistency in deep metric learning (DML) for image retrieval. Existing DML methods often result in uneven representation structures within and between classes, leading to significant variations in performance across different test classes and data distributions, measured by false accept rate (FAR) and false reject rate (FRR). To tackle this issue, the authors propose a novel variance-based metric called Operating-Point-Inconsistency-Score (OPIS) to quantify the inconsistency in threshold performance across classes. They observe a trade-off between accuracy and threshold consistency, where improving accuracy can negatively impact threshold consistency. To mitigate this trade-off, they introduce the Threshold-Consistent Margin (TCM) loss, a simple yet effective regularization technique that penalizes difficult sample pairs to encourage uniform representation structures across classes. Extensive experiments on large-scale datasets demonstrate that TCM enhances threshold consistency while maintaining or even improving accuracy, simplifying the threshold selection process in practical DML applications. The key contributions of the paper include the introduction of the OPIS metric, the identification of the accuracy-threshold consistency trade-off, and the proposal of the TCM loss as a solution to improve threshold consistency in DML. The approach outperforms state-of-the-art methods on various large-scale image retrieval benchmarks, achieving significant improvements in threshold consistency.

**Strengths:**

1. The proposed Operating-Point-Inconsistency Score (OPIS) and ϵ-OPIS provide valuable insights.
2. The experiments comparing high accuracy with high threshold consistency are objective.
3. The proposed Threshold-Consistent Margin (TCM) loss is relatively simple and easy to understand.
4. The visualization of the TCM effect is interesting.
5. The experiments are comprehensive, with detailed implementation and coverage of mainstream metric learning settings.
6. The ablation experiments are extensive, exploring margin, DML losses, different architectures, and time complexity. They also validate the proposed method against state-of-the-art approaches such as RS.

**Weaknesses:**

It is meaningful to pull the scores of positive pairs towards a fixed value and the scores of negative pairs towards another fixed value, even though it sounds simple.

Apart from that, I did not see any other weaknesses.

**Questions:**

1. Since you conducted experiments on the large-scale iNaturalist-2018 dataset, what are the differences between open-set metric learning and face recognition or re-identification (re-ID)? Can your method be applied in the field of face recognition?
2. If your method can use a single model to maintain the same threshold across multiple test sets, would it be meaningful, such as in this work[1].

[1] https://cmp.felk.cvut.cz/univ_emb/

---

> ### Author Response · Authors · 2023-11-19
> **Response to reviewer 4pGE**
>
> We thank you for thoroughly reviewing our paper.  We are delighted to learn that our proposed metric provides valuable insights, proposed method is easy to understand, and our experiments are objective and comprehensive. And we address your concerns point-by-point below.
>
> **Q1 What are the differences between open-world metric learning and face recognition or re-identification (re-ID)?**
>
> **A1** Face recognition and re-ID focus on more constrained scenarios and data domains, compared to the general image domain addressed by image retrieval tasks. Nevertheless, given the synergy between face recognition and deep metric learning, we think our development can be beneficial for face recognition applications by improving threshold consistency across various test distributions. For example, we can treat each identity as a class and apply the TCM loss across identities to encourage threshold consistency across individuals. Additionally, in our study, we also considered the ArcFace loss, which is widely-used in face recognition, on the iNaturalist-2018 dataset, as shown in Table 2 of the main paper. The results clearly demonstrate improvement in both accuracy and threshold consistency when incorporating TCM regularization into the ArcFace loss.
>
> **Q2 If your method can use a single model to maintain the same threshold across multiple test sets?**
>
> **A2** Yes, in our method, a single model can be used to test across multiple datasets and optimize for threshold consistency. We thank the reviewer for providing pointers on the dataset and setting in [1]. However, due to the presence of human faces in [Inshop dataset](https://mmlab.ie.cuhk.edu.hk/projects/DeepFashion/InShopRetrieval.html) and [Met dataset](https://cmp.felk.cvut.cz/met/), to ensure compliance with data privacy protection laws and minimize the potential infringement risks regarding personal data and privacy, we could not directly use the UnED dataset [1]. Instead, we opt for an alternative approach by merging the four datasets in our paper, namely iNaturalist-2018, Stanford Online Product, CUB-200-2011, and Cars-196, following the dataset construction methodology described in [1]. For testing, we follow [1] and equalize the number of instances per class across different domains to prevent the evaluation being dominated by majority classes with a large number of images. The following table presents the results in terms of Recall@1 and OPIS. These results highlight the efficacy of TCM regularization in cross-domain settings.
>
> Table 1. The performance of our proposed method on aforementioned dataset in Recall@1 and OPIS. The numbers in brackets represent the absolute improvement of models trained after incorporating TCM as a regularizer.
>
> |        	         |   Backbone-embedding dimension |  Recall@1 ↑	  | OPIS x 1e-3 ↓	  | Relative improvement in OPIS after incorporating TCM as a regularizer|
> |:----------------:|:-------------:|:-------------:|:---------------:|:----------------------------------------------------------------------:|
> | SmoothAP + TCM	  | ResNet50-512	 | 75.2 (↑4.1)	  | 1.55 (↓0.64) 	  |                                29.20%	                                 |
> | SmoothAP + TCM	  | ViT-B/16-512	 | 76.7 (↑5.4)	  | 2.06 (↓0.73) 	  |                                26.20%	                                 |
>
> References for Response to reviewer 4pGE:
>
> [1] Ypsilantis, Nikolaos-Antonios, et al. "Towards Universal Image Embeddings: A Large-Scale Dataset and Challenge for Generic Image Representations." *Proceedings of the IEEE/CVF International Conference on Computer Vision*. 2023.

---

> ### Author Response · Authors · 2023-11-21
> **Looking forward for further discussions**
>
> We would like to express our sincere gratitude for your diligent review of our paper. We have made the necessary improvements to our manuscript by incorporating citations of the suggested work (UnED, ICCV23) as per your recommendations. Additionally, we have conducted an experiment using a setup similar to the suggested paper and have included the corresponding results in our previous response, where our TCM loss also improves both accuracy and OPIS in this cross-domain setting. Please kindly let us know if our responses have effectively addressed your concerns. Your continued feedback is greatly appreciated, and we thank you for your valuable comments.

---

> ### Author Response · Authors · 2023-11-23
> **Additional response for Q1 on applicability of TCM in face domain**
>
> In response to your question regarding the applicability of our proposed method to the face domain, we have conducted experiments on DigiFace [2], a synthetic face dataset. Our results show improved performance in terms of both accuracy and threshold consistency when compared to ArcFace [3], as demonstrated in the results below:
>
> Table 2. The performance of ArcFace and our proposed method on DigiFace in TAR@FAR=1e-4 and OPIS.
>
> |        	        |    Backbone-embedding dimension	    |TAR@FAR=1e-4 (%) ↑ |OPIS x 1e-2 ↓ |
> |:---------------:|:---------------:|:----------------:|:------------:|
> | ArcFace + TCM	  | ResNet100-128	  |  99.54 (↑1.34)	  | 1.28(↓0.14)	 |
>
> Please feel free to reach out if you have any further questions or require additional information. Thank you for your valuable feedback and insights.
>
> References for Response to reviewer 4pGE:
>
> [2] Bae, Gwangbin, et al. "Digiface-1m: 1 million digital face images for face recognition." *Proceedings of the IEEE/CVF Winter Conference on Applications of Computer Vision*. 2023.
>
> [3] Deng, Jiankang, et al. "Arcface: Additive angular margin loss for deep face recognition." *Proceedings of the IEEE/CVF conference on computer vision and pattern recognition*. 2019.

---

### Author Response · Authors · 2023-11-19
**Response to reviewers**

We thank the reviewers (4pGE, 52BZ, 3dmW) for feedback. We are encouraged to hear that you found the OPIS metric to be “insightful” (4pGE), our TCM loss to be “novel” (52BZ), the experiments are comprehensive (4pGE, 52BZ), and our paper to be “well-written” (3dmW). We present individual responses in the subsequent sections. Additionally, we also prepared a revised manuscript (changes are highlighted in blue) and summarized modifications below.

* Added discussions and comparisons with two related works. (52BZ)
* Updated state-of-the-art results on the CUB and Cars-196 datasets. (3dmW)
* Added referenced works from all reviewers in the related works section.

---

### Comment · Area_Chair_PPrf · 2023-12-05
**Final Update**

Dear Reviewers,

Please take this chance to carefully read the rebuttal from the authors and make any final changes if necessary.

Please also respond to the authors that you have read their rebuttal, and give feedback whether their rebuttal have addressed your concerns.

Thank you,

AC

---

### Meta-Review · Area_Chair_PPrf · 2023-12-12

**Metareview:**

In this paper, the authors make the observation that threshold inconsistency presents an issue to the robustness of open-world deep metric learning. As such, the authors propose the OPIS metrics to measure the threshold inconsistency. The authors further propose threshold-consistent margin loss (TCM) as a regularization term to reduce OPIS and improve the metric learning accuracy. Experiments on multiple metric learning datasets show consistent improvements from the proposed method and compelling results.

The paper is overall well-written and the proposed method is well-motivated. A main contribution from this work is the investigation of an alternative aspect in DML beyond just accuracy, even though the practical TCM formulation is not too different from many existing large-margin designs in the rich DML/face recognition literature. Threshold consistency makes sense as a meaningful proxy to measure the system robustness and could benefit DML systems in practice. In addition, it is good to see that the improved threshold consistency also leads to improved accuracy in this work. The experimental results are comprehensive and convincing.

A main concern from the reviewers is on the missing of face verification experiments. The AC agrees to some extent in the sense that it's a pity DML works often distinguish themselves from face verification/re-id benchmarks and focus on the several classical DML benchmarks. However, the AC does not consider this a main weakness of the paper. Instead, threshold consistency has deep connections to the zero-shot robustness and generalization of the system. Similar to the introduced OOD settings in visual recognition (e.g., ImageNet-C/A/R/S), the scope and impact of paper could have benefited more had the authors pushed the boundaries more along this direction, instead of still focusing on the clean settings. In face verification, this can be related to the variation of face resolutions. In DML, this can be related to having images from different domains. This authors may consider these aspects in their future work.

Overall, this is a solid paper with good observations. The AC thus recommends acceptance to ICLR.

**Justification For Why Not Higher Score:**

As mentioned in the detailed comments section, it's a pity that the authors made good observations, but still pretty much followed the classical clean settings with DML benchmarks. There is rich literature in the DML/face recognition community regarding large-margin loss design. This paper does not differ from existing works fundamentally. In addition, DML is a relatively matured problem. This paper does not stand out in terms of interesting new applications.

**Justification For Why Not Lower Score:**

The observation and new perspective of studying DML in this work is interesting and the experimental results are solid.

---

### Decision · Program_Chairs · 2024-01-16

Accept (poster)